# META-LEARNING STRATEGIES THROUGH VALUE MAXIMIZATION IN NEURAL NETWORKS

## ABSTRACT

Biological and artificial learning agents face numerous choices about how to learn, ranging from hyperparameter selection to aspects of task distributions like curricula. Understanding how to make these 'meta-learning' choices could offer normative accounts of cognitive control functions in biological learners and improve engineered systems. Yet optimal strategies remain challenging to compute in modern deep networks due to the complexity of optimizing through the entire learning process. Here we theoretically investigate optimal strategies in a tractable setting. We present a *learning effort* framework capable of efficiently optimizing control signals on a fully normative objective: discounted cumulative performance throughout learning. We obtain computational tractability by using average dynamical equations for gradient descent, available for simple neural network architectures. Our framework accommodates a range of meta-learning and automatic curriculum learning methods in a unified normative setting. We apply this framework to investigate the effect of approximations in common meta-learning algorithms; infer aspects of optimal curricula; and compute optimal neuronal resource allocation in a continual learning setting. Across settings, we find that control effort is most beneficial when applied to easier aspects of a task early in learning; followed by sustained effort on harder aspects. Overall, the learning effort framework provides a tractable theoretical test bed to study normative benefits of interventions in a variety of learning systems, as well as a formal account of optimal cognitive control strategies over learning trajectories posited by established theories in cognitive neuroscience.

## 1 INTRODUCTION

Deploying a learning system requires making many considered decisions about hyperparameters, architectures, and dataset properties. As learning systems have grown more complex, so have these decisions about how to learn. One approach to managing this complexity is to place these decisions under the control of the agent and meta-learn them. Building on this strategy, a range of meta-learning algorithms have been developed that are capable of fast adaptation to new tasks within a distribution (Finn et al., 2017; Nichol et al., 2018), continual learning (Parisi et al., 2019), and multitasking (Crawshaw, 2020). Meta-learning methods target diverse aspects of a learning system: they can adapt hyperparameters (Franceschi et al., 2018; Baik et al., 2020; Zucchet & Sacramento, 2022); learn weight initializations well-suited to a task distribution (Finn et al., 2017; Baik et al., 2020); manage different modules or architectural components (Andreas et al., 2017); enhance exploration (Gupta et al., 2018; Liu et al., 2021); and order tasks into a suitable curriculum (Stergiadis et al., 2021; Zhang et al., 2022). While this prior work has shown that meta-learning can bring important performance benefits, algorithms are often hand-designed for a specific intervention and a large gap remains in our theoretical understanding of how meta-learning operates (see App. A).

The aim of this paper is to develop a normative framework for investigating optimal meta-strategies in biological and artificial agents. A core difficulty in computing optimal strategies is the complexity of optimizing through the learning process. To tackle this problem, we simplify the inner-loop learning dynamics using simpler tractable network models. We specifically study meta-learning dynamics in deep linear networks, which exhibit complex non-linear dynamics (Saxe et al., 2019; Braun et al., 2022). Examining this problem in a reduced setting, we derive optimal meta-learning strategies under various control designs and meta-learning scenarios. We concentrate on questions that are pertinent to the cognitive control literature, such as learning effort allocation, task switching,

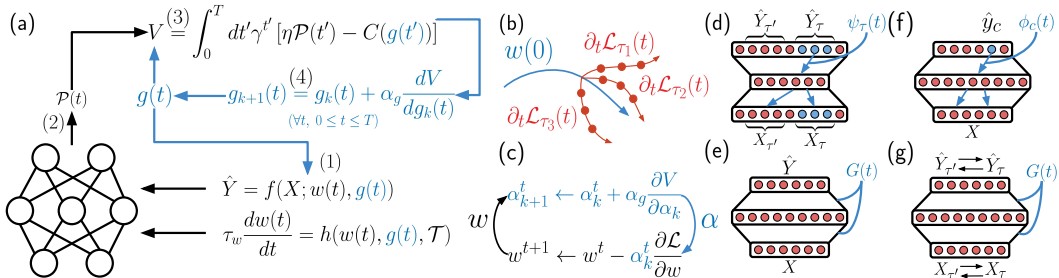

Figure 1: Learning effort framework. A neural network is under the influence of a control signal $g(t)$. This control signal is optimized iteratively by initializing $g(t)$, then: (1) Solving learning dynamics in Eq. equation 1; (2) Computing the performance $\mathcal{P}(t)$; (3) Integrating performance and control cost to compute the exact cumulative return $V$ in Eq. 2; (4) Taking the gradient of $V$ with respect to the control signal $g(t)$ and update as in Eq. 3, then go back to (1). **(b)**: Multi-step MAML. **(c)**: Learning rate optimization as in Bilevel Programming. **(d)**: Task engagement, where the control signal determines the optimal amount of *engagement* through time to multiple regression tasks. **(f)**: Category assimilation, where a model is trained to learn a classification task and can control the *engagement* on each class $c$ throughout training. **(e)**: Effort allocation, where the control signal (gain modulation of weights) is computed to maximize value throughout the learning of a single task. **(g)**: Task switching, where the gain modulation model is trained to switch tasks repeatedly and the control signal is computed throughout the switches.

and attention to multiple tasks. The Expected Value of Control Theory (EVC, (Shenhav et al., 2013; 2017; Musslick et al., 2020; Masís et al., 2021)) has proposed answers to these questions. It posits that higher-level areas in the brain perform executive functions (cognitive control) over lower-level areas to maximize the cumulative return. The framework we present is a formal and computationally tractable example of the EVC theory that takes into account the impact of the control signal on the learning dynamics (see Appendix E).

**Main contributions**

• We develop a computationally tractable *learning effort* framework[1] to study diverse and complex meta-learning interventions that normatively maximize value throughout learning.

• We fully solve learning dynamics as a function of control variables for simple models, and use this to derive efficient optimization procedures that maximize discounted performance throughout learning.

• We express meta-learning algorithms such as Model Agnostic Meta-Learning (Finn et al., 2017) and Bilevel Programming (Franceschi et al., 2018) in our framework, studying the impact of approximations on their performance.

• We compute optimized control strategies for a range of settings spanning continual learning, multi-tasking, and curriculum learning, and examine these normative strategies.

• Due to this framework's normative goal of maximizing expected return, we draw qualitative connections to phenomena in cognitive neuroscience such as task engagement, mental effort, and cognitive control (Shenhav et al., 2013; 2017; Lieder et al., 2018; Masís et al., 2021).

## 2 LEARNING EFFORT FRAMEWORK

We start by defining our framework in a general abstract way before turning to a simple example in Section 2.1. The generality of this description allows the framework to apply to a variety of different settings of interest spanning machine learning (Section 4) and cognitive control (Sections 5 and 6). Consider a learning model trained on a task $\mathcal{T}$ for a period of time $T$. Two equations define the learning model. We define the input-output mapping $f$ and learning dynamics $h$ as

$$\hat{Y} = f(X; w(t), g(t)), \quad \tau_w \frac{dw(t)}{dt} = h(w(t), g(t), \mathcal{T}) \tag{1}$$

---

[1]Anonymized Python package at `https://anonymous.4open.science/r/neuromod-6A3C/` for reproducibility.

respectively. In the first equation, $f$ is a continuously differentiable function, $X$ the input, and $\hat{Y}$ the output. Here $w(t)$ are the parameters of the learning model (e.g. weights in a neural network) during training with $T \geq t \geq 0$. We introduce $g(t)$ as an *effort signal* (or *control signal*) that crucially will be chosen by the meta-learning system. This vector of control signals can model a number of interventions in the learning system, and will be chosen to maximize cumulative learning performance. The learning dynamics equation describes the evolution of the parameters during training and is given by a differential equation over the parameters of the learning model, $h$ is a continuously differentiable function, and the evolution of the learned parameters $w(t)$ (starting at $w(0)$) may depend on the control signal and task parameters $\mathcal{T}$.

Given this setup, we can understand the control signal $g(t)$ as a free parameter that can be chosen in different ways, and which influences the network's input-output map and learning behavior. In practice, this could take the form of controlled attention or neural activity modulation. To determine how we choose $g(t)$, we define a task performance metric during the learning period $\mathcal{P}(t)$ (e.g. mean squared error during regression). Further, we assume that using the control signal $g(t)$ is costly, according to a cost function $C(g(t))$, as commonly used in control theory to describe, for instance, energy resource needed to exert control, or mental effort allocated to produced sustained engagement on a task. At any time during the learning of the task $\mathcal{T}$ we consider an instant reward rate $R(t) = \eta \mathcal{P}(t)$, where $\eta$ is a constant that converts performance on the task $\mathcal{P}(t)$ to reward/time units. We define the instant net reward rate as the difference between scaled performance and the cost of control $v(t) = R(t) - C(g(t))$. The expected return or value function at the start of training can then be written as the cumulative discounted reward from learning and performing the task from time $t = 0$ to $t = T$, with a discount factor $1 \geq \gamma > 0$,

$$V = \int_0^T dt \gamma^t v(t) = \int_0^T dt \gamma^t \left[ \eta \mathcal{P}(t) - C(g(t)) \right]. \tag{2}$$

We emphasize this value function measures performance across the whole learning period. Finally, we posit that the goal of meta-learning is to choose $g(t)$ to maximize the value function in equation 2. To find an approximately optimal $g(t)$, we take gradient steps

$$g_{k+1}(t) = g_k(t) + \alpha_g \frac{dV}{dg(t)}. \tag{3}$$

for every $0 \geq t \geq T$, $k$ being the iteration index. The optimal $g(t)$ thus depends on a complex interplay of past and future values of the control signal, and how these interact with the whole trajectory of learning. Indeed, computing the gradient in equation 3 is computationally intractable in general. In the remainder of the paper, we carefully choose learning models and settings with rich dynamics but for which we have partial analytical tractability of the learning dynamics, such that efficient computation of the full control signal through time is possible. Further details on the algorithm implemented and estimation of involved quantities can be found in App. C and D.

By appropriate choice of how $g(t)$ influences the network and learning dynamics, this general framework can accommodate a variety of possible interventions on a learning system. Some interventions correspond to other meta-learning algorithms such as Multi-Step MAML and Bilevel Programming (Fig. 1b and c). The results in subsequent sections investigate several scenarios illustrated in Fig. 1d to g. All of these experiments are variations on the influence of the control signal over the learning dynamics, keeping the rest of the framework as is.

## 2.1 SINGLE NEURON EXAMPLE

Having described the general framework, we now turn to a simple case to illustrate it, yet with complex emergent solutions. We consider a *single neuron learning model* trained on a *two-Gaussians regression task* where the control signal acts as a *weight gain modulation*. This case offers insights regarding the dependence of the optimal control signal on task parameters and learning model hyperparameters.

**Two Gaussians regression task:** A dataset of examples $i = 1, \cdots, P$ is drawn as follows: A label $y_i$ is first sampled as either $+1$ or $-1$ with probability $1/2$. The input $x_i$ is then sampled from a Gaussian $x_i \sim \mathcal{N}(y_i \cdot \mu_x, \sigma_x^2)$. The task is to predict $y_i$ from the value of $x_i$. The intrinsic difficulty of the task is controlled by how much the Gaussians overlap, controlled by the relative value of $\mu_x$ and $\sigma_x$.

**Single neuron learning model:** The input-output mapping of our single neuron model is $\hat{y}_i = x_i \cdot w(t) \left[ 1 + g(t) \right]$, $w(t)$ is our learned weight parameter, and $g(t)$ is the control signal which

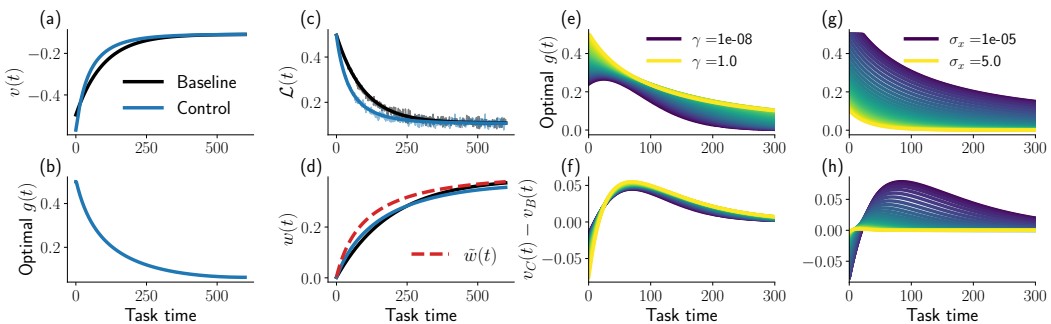

Figure 2: Results in single neuron model throughout the learning period $0 \geq t \geq T$. **(a)** Instant net reward $v(t)$. **(b)** Loss $\langle \mathcal{L}(t) \rangle$ for theoretical predictions (solid) and simulations using SGD (shaded). **(c)** Optimal control signal decreases through learning (Baseline $g(t) = 0$). **(d)** Weight $w(t)$ through learning for control and baseline case, $\tilde{w}(t) = w(t) \cdot (1 + g(t))$. Dependence of optimal control signal on task parameters. **(e)** and **(g)**: optimal $g(t)$ when varying discount factor $\gamma$ and noise level $\sigma_x$ respectively. **(f)** and **(h)**: Difference between instant net rewards $v(t)$ between control and baseline when varying $\gamma$ and $\sigma_x$ respectively. Longer time horizons and less noisy tasks recruit more control.

acts as a multiplicative gain. The learning dynamics of $w(t)$ are given by gradient descent on the loss function $\mathcal{L} = \frac{1}{2}(y_i - \hat{y}_i)^2 + \frac{\lambda}{2}w(t)^2$. Taking the gradient flow limit (small learning rate (Saxe et al., 2019; Elkabetz & Cohen, 2021)), we find average learning dynamics for the weight described by

$$\tau_w \frac{dw}{dt} = -\left\langle \frac{\partial \mathcal{L}}{\partial w} \right\rangle = \mu_x \tilde{g}(t) - w(t) \left( \langle x^2 \rangle \tilde{g}^2(t) + \lambda \right) \tag{4}$$

where $\tilde{g}(t) = 1 + g(t)$, $\langle \cdot \rangle$ denotes expectation over the data distribution, and $\tau_w$ is the learning time scale of the weight. This gradient depends on $g(t)$, making the learning dynamics of $w(t)$ dependent on the control signal. For the single neuron model, we can find a closed form expression for $w(t)$ as a function of the control signal $g(t)$, giving us an expression for $\langle \mathcal{L}(t) \rangle$ as well (see App. G.1). This tractability allows us to compute average dynamics and the necessary gradient efficiently.

**Control signal optimization**: As a performance and control measure for this model we use $\mathcal{P}(t) = -\langle \mathcal{L}(t) \rangle$, $C(g(t)) = \beta g(t)^2$ respectively, meaning smaller loss leads to better performance and exerting control has a cost that is monotonic in the control signal magnitude with cost per unit of control $\beta$. Note that if $g(t) = 0$ for all $T \geq t \geq 0$, then $C(g(t)) = 0$, and $\hat{y}_i = x_i \cdot w(t)$, which means that the weight is learned purely by gradient descent with no influence from the control signal. This is the *Baseline* model. Having $\mathcal{P}(t)$ and $C(g(t))$, we can compute the value function in equation 2 and find the optimal $g(t)$ by gradient ascent following equation 3 (algorithm described in App. D). In essence, this setting considers a simple learning scenario in which an agent can adjust the gain of the weights in a neural network that otherwise learns via gradient descent.

**Results:** In Fig. 2a, we show the difference in instant net reward $v(t)$ for the baseline ($g(t) = 0$ for every $t$) and the control case (optimizing $g(t)$). The optimal meta-learning strategy that maximizes expected return in equation 2 invests more control at the start (Fig. 2b) of the learning period at the cost of some instant reward, with the result of faster learning (demonstrated in the lower loss for the control case in Fig. 2c). The control signal $g(t)$ influences the instant net reward rate both at present $t$ and future $t' > t$. The instant change in net reward rate $v(t)$ will be caused by both the instant change on $\tilde{w}(t) = w(t) \cdot (1 + g(t))$ (Fig. 2d) and $C(g(t))$, making the effective weight $\tilde{w}(t)$ closer to the solution at early stages. As expected, increasing the discount factor $\gamma$ leads to higher levels of control, since future net reward will contribute more to the cumulative expected return, compensating the cost of increasing $g(t)$ (Fig. 2e,f). Increasing the intrinsic noise of the task $\sigma_x$ reduces the overall optimal control (Fig. 2g,h). Because it is not possible to overcome this noise, the use of control will generate a cost that cannot be compensated by boosting learning. This inter-temporal choice of allocating effort based on the prospect of future reward has been widely studied in psychology and neuroscience (Masís et al., 2021; Keidel et al., 2021; Frömer et al., 2021; Masís et al., 2023) (App. A), and naturally arises from maximizing the discounted cumulative performance in Eq. 2. For more parameter variations see Fig. 10 in App. K.1.

## 3 BASELINE DEEP LINEAR NETWORKS AND DATASETS

We now generalize the single neuron approach to more complex neural networks. In the case of a two-layer linear neural network, the corresponding input-output mapping in Eq. equation 1 is $\hat{Y} = W_2(t)W_1(t)X$, where $X \in \mathbb{R}^I$, $\hat{Y} \in \mathbb{R}^O$, $W_1(t) \in \mathbb{R}^{H \times I}$ and $W_2(t) \in \mathbb{R}^{O \times H}$ are the first and second layer weights. Training a two-layer network to minimize MSE with weight regularization and taking gradient flow limit yields the learning dynamics equations (Saxe et al., 2019; Braun et al., 2022)

$$\tau_w \frac{dW_1}{dt} = W_2^T \left( \Sigma_{xy}^T - W_2 W_1 \Sigma_x \right) - \lambda W_1, \quad \tau_w \frac{dW_2}{dt} = \left( \Sigma_{xy}^T - W_2 W_1 \Sigma_x \right) W_1^T - \lambda W_2 \quad (5)$$

where $\Sigma_{xy}^T = \langle XY^T \rangle$, $\Sigma_x^T = \langle XX^T \rangle$, $\tau_w$ is a learning time-scale of the weights, and $\lambda$ controls the weight regularization (see App. H). Learning is completely defined by the initial weights $W_1(0)$, $W_2(0)$, the task at hand and the hyperparameters, it follows non-linear dynamics due to weight coupling and a non-convex loss landscape while keeping computational tractability. With the general framework and tractable models in hand, we now turn to probe the behavior in a variety of settings. First, in Section 4, we draw out implications for standard meta-learning algorithms. Next, in Section 5 we turn to aspects of curriculum learning and the choice of which tasks to engage with. Finally, in Section 6 we study control interventions in the form of gain modulation throughout a network, of relevance to theories in neuroscience. In all of these sections, we compose meta-learning tasks from a base set of three datasets: (1) **Correlated Gaussian** regression, (2) **Semantic Tasks** with hierarchical concepts, and (3) **MNIST** (Details in App. J), from which we can determine the statistics needed to compute the learning dynamics (e.g. $\Sigma_x$ and $\Sigma_{xy}$).

## 4 RELATION TO META-LEARNING ALGORITHMS IN MACHINE LEARNING

The normative objective in Eq. equation 2 and the way it is maximized through gradient steps on the control signal $g(t)$ can describe other meta-learning algorithms. Here we show the connections to two well-established algorithms, Model Agnostic Meta-Learning (MAML (Finn et al., 2017)) and Bilevel Programming (Franceschi et al., 2018).

MAML is an instance of our framework where the initial weights $W_1(0)$ and $W_2(0)$ in our deep linear network *are* the control signal $g(t)$. By defining the performance as the average loss per task indexed by $\tau$, $\mathcal{P}(t) = \sum_\tau \langle \mathcal{L}_\tau(t) \rangle$, this becomes the meta-objective in MAML when considering only one step ahead in the value function, this is $V_{\text{MAML}} = \mathcal{P}(\delta t)$ with $\delta t$ being the time after one gradient update on $g(t)$ (See App F.1). Our framework can also optimize performance after multiple gradient

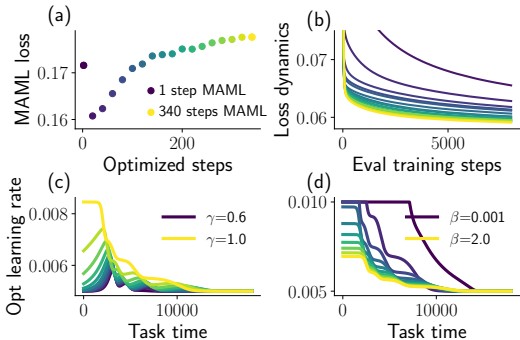

Figure 3: **(a)**: Single step MAML loss $V = \mathcal{P}(\delta t)$ when considering more steps in the learning dynamics. **(b)**: Resulting learning dynamics from initial parameters found with Multi-Step MAML. **(c) and (d)**: Optimal learning rate when varying discount factor $\gamma$ and cost coefficient $\beta$.

steps, therefore obtaining Multi-Step MAML in a computationally tractable setting (Fig. 1b). We used the two-layer linear network (Section 3) and a set of 5 binary regression tasks with different pairs of digits from MNIST (App. F.1) to simulate Multi-Step MAML. Results in Fig. 3a and b show that the standard MAML loss $V_{\text{MAML}}$ changes depending on how many steps ahead are considered during the initial weights optimization. $V_{\text{MAML}}$ decreases when considering a few steps ahead, increasing the capacity to optimize the dynamics. After a certain number of steps considered in the optimization, $V_{\text{MAML}}$ increases, sacrificing immediate performance to optimize the dynamics in steps further away, as shown in Fig. 3b. These multi-step results are only possible due to the tractability of our setting. We see that one-step MAML can substantially underperform Multi-Step MAML.

We also optimized hyperparameters of the network throughout training. Bilevel Programming optimization can compute this, with the main distinction being the reverse-hypergradient method used to update the meta-parameters (control signal) (Franceschi et al., 2017; 2018). We extend

these methods by adding features with intuitive meaning under our normative frameworks, such as the discount factor $\gamma$ and control cost $C$ (App. F.2). We optimized the learning rate throughout time to maximize the cumulative reward in equation 2, and varied $\gamma$ and $\beta$ as in the single neuron example to illustrate the normative meaning in a hyperparameter optimization context (Fig. 1c). We observed qualitatively similar behavior as in the single neuron model, longer time horizons and less cost of increasing the learning rate recruit more control. We provide an additional example of meta-learning the learning rule using our framework in App. F.3. Our work provides further utility to these meta-learning algorithms by interpreting them under a normative value-based framework.

## 5 ENGAGEMENT MODULATION

Next we turn to the question of which tasks among many to engage with over time. We provide the model control over its *engagement* on a set of available tasks, or class in a classification problem during learning. Selecting the optimal control signal in this setting involves improving multi-task capabilities and estimating optimal curriculum. Consider a set of $N_\tau$ datasets, and a loss function $\mathcal{L}(\hat{Y}_\tau, Y_\tau)$, where $\hat{Y}_\tau$ is the estimation of a model and $Y_\tau$ is the required target for dataset $\tau$. The average loss for a set of datasets is $\mathcal{L} = \sum_{\tau=1}^{N_\tau} \mathcal{L}(\hat{Y}_\tau, Y_\tau) + \mathcal{R}(W)$ which is used to measure the performance $\mathcal{P}(t) = -\langle \mathcal{L}(t) \rangle$ only, and we assume that weights are updated via gradient descent on the auxiliary loss $\mathcal{L}_{\text{aux}} = \sum_{\tau=1}^{N_\tau} \psi_\tau(t)\mathcal{L}(\hat{Y}_\tau, Y_\tau) + \mathcal{R}(W)$, where $\psi_\tau(t)$ are control signals we call *engagement coefficients*, and $\mathcal{R}(W)$ is a weight decay regularizer. Assuming the network receives inputs from all of the datasets at the same time (concatenated in $X$) and has specific outputs allocated to each dataset (concatenated in $Y$) as schematized in Fig. 1d, we can derive the learning dynamics equations for the weights as a function of $\psi_\tau(t)$ giving

$$\tau_w \frac{dW_1}{dt} = \sum_\tau \psi_\tau(t) W_{2\tau}^T \left( \Sigma_{xy\tau}^T - W_{2\tau} W_1 \Sigma_x \right) - \lambda W_1,$$

$$\tau_w \frac{dW_2}{dt} = \sum_\tau \psi_\tau(t) \left( \Sigma_{xy\tau}^T - W_{2\tau} W_1 \Sigma_x \right) W_1^T - \lambda W_2, \tag{6}$$

where $W_{2\tau}$ denotes the weights of the neurons for the output to dataset $\tau$ and $\Sigma_{xy\tau}$ is $\langle XY_\tau^T \rangle$, both padded with zeros to preserve dimension (see App. H.2). Each of the $\psi_\tau(t)$ modulates the amount of learning of each dataset. The auxiliary loss to get a learning dynamics is to avoid the trivial solution of $\psi_\tau = 0$ to minimize the loss. We can find the optimal $\psi_\tau(t)$ throughout learning by computing $\mathcal{P}(t)$, using $C(\psi(t)) = \beta \|\mu_\psi - \bar{\psi}(t)\|^2$ ($\bar{\psi} = (\psi_1(t), \psi_2(t), ...)$), then taking gradient steps on $\psi_\tau(t)$ to maximize $V$. Taking $\mu_\psi = 0$ means that to learn a dataset $\tau$ ($\psi_\tau(t) > 0$) the agent must pay a cost. We call this case *active engagement*. For $\mu_\psi = 1$, the agent must pay a cost to increase or suppress the learning signal from a specific dataset relative to a baseline. We call this case *attentive engagement*. In these cases, each of the elements in $\bar{\psi}(t)$ are forced to stay in a certain range independently. Finally, we can force $\bar{\psi}(t)$ to be of a fixed norm by making the cost $C(\bar{\psi}(t)) = \beta \left( \|\bar{\psi}(t)\|^2 - \Psi \right)^2$, such that there is a fixed overall amount of engagement to distribute. We call this case *vector engagement*. For category engagement, which is focusing on particular subclasses in a classification problem, a similar set of equations can be derived (see App. H.3), where the engagement on class $c$ through learning is denoted by $\phi_c(t)$ (Fig. 1f). The meta-learning tasks used to train this model are the following.

**Task engagement**: Given a set of $N_\tau$ datasets, and a total training period of $T$, we trained the engagement modulation model described in Section 5. The idea of this task is to estimate the optimal learning curriculum (order of datasets presented in the neural network training) that maximizes expected return $V$ during the time period $T$. In this task, three binary MNIST classification datasets were used, specifically the digits $(0, 1)$, $(7, 1)$ and $(8, 9)$ ordered by difficulty (easier to harder according to linear separability, see App. H.2).

**Category engagement**: For a classification task, there might be a better set of classes to learn during different stages of training. We trained the category engagement modulation model (described in Section 5 and App. H.3) to estimate the optimal *engagement* or *attention* to each of the categories in a classification task (Semantic and MNIST datasets) through learning. In addition, we trained the gain modulation model (next Section) in this same setting using a *neuron basis* (see App. H.1).

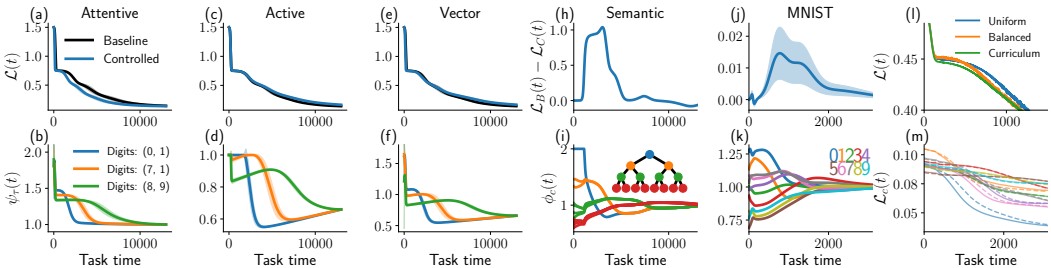

Figure 4: Results for task engagement experiment. **(a)**, **(c)** and **(e)**: $\mathcal{L}(t)$ for baseline and control case for Attentive, Active and Vector engagement. **(b)**, **(d)** and **(f)**: Engagement coefficients $\psi_\tau(t)$ for each of the binary classification tasks Attentive, Active and Vector engagement. Mean and standard deviations from 5 independent trainings. **(h)** and **(j)**: Results for category engagement task, improvement in the loss function when using control for MNIST and Semantic dataset respectively. **(i)** and **(k)**: Optimal category engagement coefficients for MNIST and Semantic datasets. **(l)**: Class proportion experiment. **Uniform**: Loss when using uniform distribution for the abundance of classes in each batch. **Balanced**: Loss on a balanced batch, but using the inferred curriculum of classes in the batch to train. **Curriculum**: Loss on curriculum batch when using the curriculum. **(m)**: Loss per class using control (solid lines) and baseline (dashed lines).

## 5.1 RESULTS

**Task engagement**: We simultaneously gave the neural network inputs and targets for three datasets, as described in Section 5, each of them a different binary regression problem from MNIST. Each dataset used was chosen to vary on the level of difficulty to learn: the pair of numbers (0, 1) is easier to classify than (7, 1) and (8, 9) (based on the lowest loss achievable with linear regression in App. K.4). We computed the engagement coefficients $\psi_\tau(t)$, one per dataset, that maximizes the expected return in Eq. 2. Learning curves and the evolution of engagement coefficients are depicted in Fig. 4; the baseline case corresponds to simultaneous training on all datasets at the same time ($\psi_\tau(t) = 1$ and $C(\psi(t)) = 0$). In the *attentive engagement* agent, where $\mu_\psi(t) = 1$ (shown in Fig. 4a and 4b), the agent just needs to pay a cost to either amplify or suppress engagement on a dataset. In this setting, the agents amplify the engagement of all of the datasets, effectively increasing the learning rate per dataset, and achieving a lower $\mathcal{L}_C(t)$ compared to $\mathcal{L}_B(t)$. The order of learning each of the datasets goes from easier to harder, it is in the same order as in the *active engagement* and *vector engagement*, and none of the datasets are engaged with $\psi_\tau(t) < 1$, presumably to avoid forgetting of early amplified datasets. In the case of *active engagement*, where $\mu_\psi = 0$ in $C(\psi(t)) = \beta\|\mu_\psi - \bar{\psi}(t)\|^2$ (shown in Fig. 4c and 4d), the agent must pay a cost to learn any of the tasks ($\psi_\tau(t) > 0$). By distributing the learning between the tasks, the agent is capable of reaching $\mathcal{L}_C(t)$ close to $\mathcal{L}_B(t)$ as shown in the top panel of Fig. 4, without the need of fully engaging on all of the datasets at every time step. None of these datasets are fully disengaged at any point, possibly to avoid catastrophic forgetting of datasets previously engaged during training. The engagement coefficients in the *vector* case behave similarly. Since the control signal in this case is forced to keep a constant size of $\Psi$, the agent is not able to fully engage in all of the datasets, and distributes this *attention* resource on each dataset from easier to harder, as in the *active* case. The meta-learning strategy found in our setting of keep re-visiting previous tasks to keep performance is well studied in psychology (Ericsson & Harwell, 2019; Eglington & Pavlik Jr, 2020), and it is also related to memory *replay* theories as a value-based mechanism that avoids catastrophic forgetting (Mattar & Daw, 2018; Agrawal et al., 2022).

**Category engagement**: In some classification tasks, it might be better to learn some categories first and others later during training. We trained the engagement modulation model to control *engagement* or *attention* on categories of a classification dataset. In Fig. 4 we show the results of this model trained on the Semantic dataset, and MNIST dataset classifying all digits. The engagement coefficients $\phi_c(t)$ describe the focus on class $c$ in the classification problem, which basically scales the error signal for that specific class through training (see App. H.3). Fig. 4h and 4j show the improvement in the loss when optimizing for categoric assimilation coefficients for both datasets. Fig. 4i and 4k depict the engagement coefficients per class $\phi_c(t)$. In the Semantic task, the basis coefficients are clustered depending on the level of the hierarchy for the respective output. Higher coefficients are spent on categories in higher levels of the hierarchy, as well as earlier during learning. Because we kept $\beta$ high

for this experiment ($\beta = 5.0$), the cost of deviating from a control vector of size $C$ is high (where $C$ is the number of classes); therefore the amplification of engagement in some categories goes along with suppression for other categories to keep the control with constant size $C$. For the MNIST dataset, each $\phi_c(t)$ corresponds to a specific digit, and the order of assimilation that maximizes value shows a consistent order of digits among different runs, being ordered as $(0, 1, 7, 6, 4, 2, 3, 9, 8, 5)$, which is roughly the same as the average linear separation per digit (see App. K.4). As in the task engagement results, we found that it is optimal to assimilate easier elements first, allocating higher $\phi_c(t)$ and more concentrated in the early stages of learning. More difficult categories are assimilated later, allocating a smaller maximum $\phi_c(t)$ compared to easier classes, but with sustained engage over time. The benefits of learning from easier to harder aspects of tasks have been shown in cognitive science (Krueger & Dayan, 2009; Wilson et al., 2019) and machine learning (Parisi et al., 2019; Saglietti et al., 2022; Zhang et al., 2022), and we are able to reproduce this finding in the task engagement and category engagement experiments within our normative framework. The engagement level per class amplifies the error signal of learning a particular class through time, which can be roughly controlled by modifying the proportion of classes in the batch through training. To show this, we trained the baseline network on MNIST (no control, only backpropagation), and used $\phi_c(t)$ to modify the proportion of classes in the batch throughout the training (App. H.3). This gives a better curriculum than sampling each class uniformly to populate the batch, as shown in Fig. 4l and 4m.

## 6 GAIN MODULATION

Motivated by studies of neuromodulation (Lindsay & Miller, 2018; Ferguson & Cardin, 2020), we finally address a neuroscience inspired model (Shenhav et al., 2013; 2017) where the *learning effort control signals* $G_1(t) \in \mathbb{R}^{H \times I}$ and $G_2(t) \in \mathbb{R}^{O \times H}$ modulate the gain of each layers weights as $\tilde{W}_i(t) = (\mathbb{1} + G_i(t)) \circ W_i(t) = \tilde{G}_i(t) \circ W_i(t)$ where $\circ$ denotes element-wise multiplication. This control signal will modify the input-output mapping of the network to $\hat{Y} = \tilde{W}_2(t)\tilde{W}_1(t)X$. Given the control signals, we assume the weights are learned using gradient descent, yielding the learning dynamics equations

$$\tau_w \frac{dW_1}{dt} = \left( \tilde{W}_2^T \Sigma_{xy}^T \right) \circ \tilde{G}_1 - \left( \tilde{W}_2^T \tilde{W}_2 \tilde{W}_1 \Sigma_x \right) \circ \tilde{G}_1 - \lambda W_1,$$

$$\tau_w \frac{dW_2}{dt} = \left( \Sigma_{xy}^T \tilde{W}_1^T \right) \circ \tilde{G}_2 - \left( \tilde{W}_2 \tilde{W}_1 \Sigma_x \tilde{W}_1^T \right) \circ \tilde{G}_2 - \lambda W_2. \tag{7}$$

The control signal $G_i(t)$ effect is *similar* to a time-varying learning rate, except (1) it is weight specific (i.e. with coupling between the elements of the control matrix), (2) it does not change the weight decay rate which is originally controlled by $\lambda$ and $\tau_w$, and (3) $G_i(t)$ also changes the input-output mapping. Solving the learning dynamics gives $\mathcal{P}(t) = - \langle L(t) \rangle$, using $C(G(t)) = \exp \left( \beta \left( \|G_1(t)\|_F^2 + \|G_2(t)\|_F^2 \right) \right) - 1$, to then estimate $dV/dG_i(t)$ as in App. C, and find the control trajectory that maximizes cumulative reward in Eq. 2 (More details in App. H.1, we provide an exact solution of the learning dynamics in a single-layer network given a control signal $G(t)$ in App. G.2). In addition, we simulated this model in a non-linear network using approximations (see App. I). The meta-learning tasks used to train this model are the following.

**Effort Allocation**: We train the gain modulation model separately on each of the three datasets for a time period of $T$, and estimate the control signal that maximizes the expected return $V$ in Eq. 2.

**Task Switch**: We defined two different Gaussian datasets (App. L). We sequentially train the network on each dataset for a time period $T_s$. The expected reward $V$ is computed for the whole training period $T > T_s$ of the gain modulation model, and maximized through gradient updates on $G_i(t)$.

### 6.1 RESULTS

**Effort Allocation**: This setting is similar to the single neuron setting of Section 2.1, but with a two layers network instead of just one neuron, where every weight in the network has its own gain signal as described in Eq. 7 and schematized in Fig. 1e. The results of the baseline training and controlled training using gain modulation are presented in Fig. 5. In the gain modulation model, we can see the same qualitative behavior as in the single-neuron model when varying parameters of the learning model and control optimization. The control signal that maximizes expected return reduces the instant net reward rate by the use of control in the early stages of learning, to get better performance

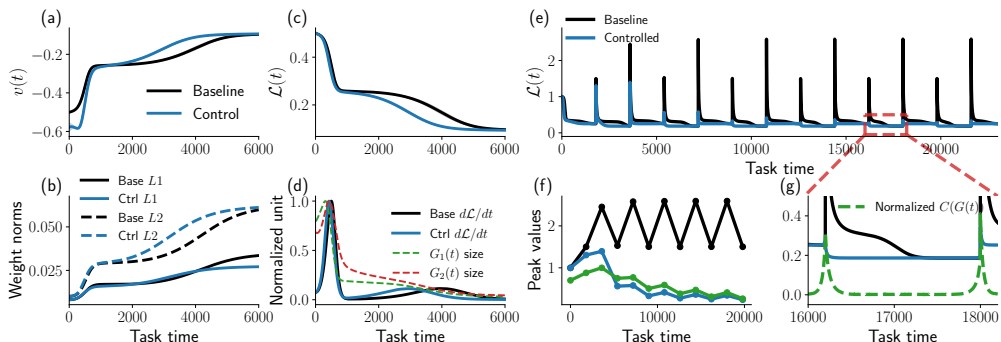

Figure 5: Results of the gain modulation model trained on an MNIST classification task. **(a)**: Instant net reward $v(t)$, baseline vs controlled. **(b)**: L1 and L2 norms of the weights. **(c)**: Loss $\mathcal{L}(t)$ throughout learning. **(d)**: normalized $d_t\mathcal{L}(t)$, and normalized L2 norm of the control signal $G_1(t)$ and $G_2(t)$. **(e)**: Results on the task switch meta-task. Comparison of $\mathcal{L}(t)$ for the baseline and control case. **(f)**: Values of $\mathcal{L}(t)$ at switch times, along with the normalized cost of control $C(t)$ at switch times (green line). **(g)**: Zoom of $\mathcal{L}(t)$ in the top panel, along with the normalized cost of control.

later as shown in the lower loss for the controlled case (Fig. 5a and b). Through both optimizing the learning and minimizing $C(G(t))$ at the same time, the gain modulation is not only more efficient by getting a more sparse solution ($L_1$ norm in Fig. 5b), using fewer weights than when no control is used, it also learns faster (Fig. 5c, more details in App. K.2). There are times during learning when it is more effective to apply control. As can be seen in the $L_2$ norm of the control matrices $G_1(t)$ and $G_2(t)$, and the absolute value of the time derivative of the loss $d_t\mathcal{L}(t) = |d\mathcal{L}(t)/dt|$ for the baseline and control case (Fig. 5d), the control signal is larger early in training and near the stages of learning when the increase in performance ($d_t\mathcal{L}(t)$) is larger (Fig. 5a). The control signal shifts the peaks in $d_t\mathcal{L}(t)$ earlier in learning, leading to better performance and higher reward earlier, compensating for the momentarily increased cost of control. Similar results are obtained when training on the other two datasets (see Fig. 13 and 14 in App. K.2). Neuromodulators are known to be involved in high-level executive tasks such as engagement in learning (Shenhav et al., 2013; 2017; Lieder et al., 2018; Grossman et al., 2022), and some of them are believed to act as gain modulation (Lindsay et al., 2020; Ferguson & Cardin, 2020) (App. A). We provide a testable and tractable setting where neuromodulators could manage performing and learning tasks to maximize cumulative reward.

**Task Switch**: The task is schematized in Fig. 1g. In Fig. 5e, each peak in the loss is a task switch (every 1800 time steps), and as expected, the baseline loss $\mathcal{L}_B(t)$ is higher than the loss with control $\mathcal{L}_C(t)$ almost at every point throughout learning. After each switch, the control signal manages to iteratively drive the learning dynamics to places in parameter space $W$ where each switch is less costly (Fig. 5f). Since the linear network is over-parametrized, the drive to adjust for the next task can be done without meaningfully changing the solution for the current task. The control signal starts acting before the switch (Fig. 5g) to amortize the loss peak at the time of the switch, and to speed up the approach of the weight to the solution, skipping the plateau in the loss. In addition, the sparsity of the weights is higher compared to the baseline case, the cost of using control to switch is transferred to the size of the weights , making it easier to move the effective weight $\tilde{W}(t)$ by a large amount when changing $G(t)$ (See App. K.3). This setting poses meta-learning and gain modulation as a neural implementation of task/context switching in real scenarios (Puigbò et al. 2020; Ben-Iwhiwhu et al. 2022).

# 7 DISCUSSION

We present a flexible computationally tractable *learning effort* framework to study optimal meta-learning with neural network dynamics in a variety of settings. Our framework optimizes control signals on a fully normative objective: discounted cumulative performance throughout learning. Our goal is to provide formal underpinnings for cognitive control theories in neuroscience, and aid the evaluation of possible interventions in engineered systems (see App. B). While a limitation of this work is its use of linear network models, we study the dynamics of non-linear networks in App. K.6, obtaining similar results as the ones found using linear networks. We hope our work will contribute to a greater understanding of how agents should act to maximize their learning abilities.

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

## A    FURTHER RELATED WORK

In recent years, several meta-learning algorithms have been proposed to solve a variety of meta-learning tasks, such as fast-adaptation (Finn et al., 2017; Nichol et al., 2018), continual-learning (Parisi et al., 2019), and multi-tasking (Crawshaw, 2020; Sagiv et al., 2020; Ravi et al., 2021; Musslick et al., 2020). Because these tasks have different goals, the specific design of the meta-learning algorithm used to solve each task differs.

One popular application for meta-learning algorithms is reinforcement learning (RL). RL agents use a policy to choose actions to maximize the expected return. The policy is usually based on a value function (or action-value), linking particular actions to values, that agents learn to estimate through experience (Mnih et al., 2015; Wang et al., 2017). For the agent to act optimally, the policy requires a good estimation of the value function. For single tasks, this is typically not hard, but agents struggle when they must solve more than one task. To aid this difficulty, researchers implement meta-learning strategies, such as enhancing exploration (Gupta et al., 2018; Liu et al., 2021), re-using experiences through a memory buffer (Ritter et al., 2018), exposing the agent to a large number of tasks from a task distribution (Wang et al., 2017; Team et al., 2021), and choosing the order of those tasks carefully (Stergiadis et al., 2021; Zhang et al., 2022) with the hope that the agent performs well on each of these tasks in the task distribution. Indeed, several techniques improve the performance of reinforcement learning agents (Hessel et al., 2017; Obando-Ceron & Castro, 2021; Kanervisto et al., 2021), but there are two main issues with these approaches. First, these techniques are usually designed manually and specifically for the tasks at hand. Second, how the learning dynamics depend on these techniques remains unclear. The combined effects of the agent-environment interaction dynamics and the value estimation during training make analyzing learning dynamics on these models remarkably challenging. A technique that could autonomously generate a meta-learning strategy *a priori* by leveraging analysis of the learning dynamics would address these issues and potentially improve the understanding, performance, and flexibility of these types of models.

Learning dynamics has been widely studied in the context of neural network training, where the goal in these cases is to minimize a loss function. In particular, deep linear networks (Saxe et al., 2019; Zenke et al., 2017; Li & Sompolinsky, 2021; Braun et al., 2022), gated linear networks (Saxe et al., 2022; Li & Sompolinsky, 2022), have been useful to analyze learning dynamics, due to their mathematical tractability and still complex (non-linear) learning dynamics. Having access to the learning dynamics allows us to test the learning system under different conditions (tasks, architectures, hyperparameters, etc) and draw conclusions, either from mathematical analysis or simulations on how these conditions affect learning during the training period. Further techniques to describe learning dynamics exist, which have their drawbacks in terms of mathematical and computational tractability, or required limits in input or hidden dimensionality to obtain closed-form differential equations describing the dynamics. Some of these frameworks are the teacher-student settings (Goldt et al., 2019; Ye & Bors, 2022), and mean-field theory for neural networks (Mignacco et al., 2020; Bordelon & Pehlevan, 2022), recently applied to temporal difference learning (Bordelon et al., 2023). These methods present a promising direction to extend our framework to non-linear networks and reinforcement learning dynamics.

In this work, we propose a new framework called *learning effort*, where we combine the goal of maximizing value, with neural network learning dynamics, by making the regression loss during training proportional to the reward of the learning system (in this case, a neural network). This has several advantages. First, in practice, this choice makes the problem of estimating value equivalent to estimating the learning dynamics. Because the loss function during training can be obtained from fully solving the learning dynamics, then the reward throughout training is also solved (similar to Zenke et al. 2017). Second, taking advantage of the partial tractability of linear networks, and approximating non-linear network dynamical equations, we are able to draw conclusions on how some parameters interact with the learning dynamics when maximizing the value. Using this framework, we define a control signal, that at a cost, can modify the learning dynamics to maximize value during training. Furthermore, any network parameter that is not subject to the learning dynamics could be chosen as this control signal to maximize value. Previous work has addressed these questions by assuming a learning trajectory, where the functional form is fixed, hence not considering possible changes in the trajectory due to the time-varying control (Son & Sethi, 2006; 2010). Other work along these lines improves this by training complex learning systems and learning the value function to estimate the optimal control signal (Musslick et al.; Lieder et al., 2018). Some meta-learning algorithms such as

MAML (Finn et al., 2017) and bilevel optimization methods (Franceschi et al., 2018; Andreas et al., 2017) can be encapsulated under our framework as explained in Section 4 in the main manuscript. For further meta-learning-related work and formal links to our framework see Appendix F.

We used this framework to investigate different kinds of intervention of the control signal in the learning process, finding what are optimal strategies when facing a meta-learning task, including when the use of the control signal is costly. Using this setting, we can ask questions such as how to optimally allocate control to speed up learning while minimizing the use of it, how to train the network to switch tasks quickly, or what is the optimal level of attention to a set of tasks, or even if it is worth to learn a specific task given the cost of engaging on learning it, all by maximizing value during learning. Related work has used similar bi-level optimization, but with different loss functions on each optimization level (Zucchet & Sacramento, 2022). We provide the implementation for the work presented here in `https://anonymous.4open.science/r/neuromod-6A3C`.

We further suggest that this same framework could be useful to analyze phenomena in cognitive neuroscience, and that there is a correspondence of our *learning effort* framework and the Expected Value of Control Theory (Shenhav et al., 2013; Musslick et al., 2020; Masís et al., 2021). We provide an extended discussion referring to the links to cognitive neuroscience literature in App. B.

## B    EXTENDED DISCUSSION

In this extended discussion, we further consider emerging principles from our results and the utility of our framework for theories of cognitive control in neuroscience. This learning effort framework is flexible enough to pose different questions about meta-learning strategies by slightly varying the original setting, and keeping the conceptual framework fixed. The main connection is through the expected value of control theory (Shenhav et al., 2013; Keidel et al., 2021; Frömer et al., 2021; Masís et al., 2023), formally explained in App. E. It is also proposed that the dorsal anterior cingulate cortex (dACC) is involved in the integration and computation of most of the quantities needed in the EVC theory, which is also supported by some neuroimaging experiments (Botvinick et al., 2001) and it is consisting with other theories (Botvinick & Cohen, 2014). **With our framework, it is possible to compute optimal solutions to the EVC problem efficiently, with the additional feature of considering the impact of control in complex learning dynamics throughout the entire training.**

One of the emergent solutions we found is that, **it is generally better to allocate resources in the early stages of learning to harvest higher rewards later**. This inter-temporal choice of allocating effort based on the prospect of future reward has been widely studied in psychology and neuroscience (Masís et al., 2021; Keidel et al., 2021; Frömer et al., 2021; Masís et al., 2023) (App. A). One example is (Masís et al., 2023) where it is observed that rats somehow manage their reaction times to improve learning speed on a classification task. Longer reaction times in leary stages of learning lead to less instant reward rate, but speed up the performance through sessions, and get a higher cumulated reward throughout the entire learning process (see Figure 5, panel (e), (f) and (g) from (Masís et al., 2023). This looks qualitatively similar to our single neuron example, gain modulation experiments, and task engagement experiments). This phenomenon is a form of a dynamic *marshmallow test* experiment (Mischel, 2014), where kids face a decision of eating a marshmallow now, or waiting 20 minutes more to get a second marshmallow, which needs some amount of self-control. The main feature of our setting is the effect the control signal in this decision affects learning dynamics through the task, which adds an extra level of abstraction on meta-cognition about the learning capabilities of the agents taking the decision (Son & Sethi, 2006; 2010; Musslick et al., 2020).

Another emergent solution is that **easier tasks/categories should get more resources in the early stages of learning compared to the harder ones, which sustained effort through training**. As mentioned in the main text, this strategy has been shown in cognitive science (Krueger & Dayan, 2009; Wilson et al., 2019) and machine learning (Parisi et al., 2019; Saglietti et al., 2022; Zhang et al., 2022). One possible reason for not disengaging in a task or category at any point could be to avoid catastrophic forgetting or interference between tasks/categories, since they are all sharing the hidden layer. We hypothesize this phenomenon can be posed as a memory replay system, and we provide a framework where this is a value-based mechanism, as indicated by other work (Mattar & Daw, 2018; Agrawal et al., 2022).

In addition, the fact that **increasing the cost of control reduces control allocation** has also been observed in cognitive science and it is denoted as avoidance of cognitive demand (Kool et al., 2010; Kool & Botvinick, 2014; Westbrook & Braver, 2015). For instance, strategies that require high cognitive demand (our cost term $C$ in our framework) will be naturally avoided even if this strategy is optimal to solve the task. However, as shown in Experiment number 6 in (Kool et al., 2010), the subjects will engage in the optimal high cognitive demand strategy if they get paid to solve the task efficiently. This is, in an abstract way, increasing $\eta$ in our framework, which is equivalent to decrease $\beta$. We observe higher engagement in control when $\beta$ is decreased as shown in this experiment, and it has been described in (Kool & Botvinick, 2014) as a labor leisure trade-off. Our framework can potentially explain cognitive fatigue and boredom from a normative perspective of maximizing value, which has been theorized in (Agrawal et al., 2022).

Possible neural implementations of a control signal in our framework are gain modulation (Section 6) and error modulation (named engagement modulation in our work, Section 5). Neuromodulators are known to be involved in high-level executive tasks such as engagement in learning (Shenhav et al., 2013; 2017; Lieder et al., 2018; Grossman et al., 2022), and some of them act as gain modulation (Lindsay et al., 2020; Ferguson & Cardin, 2020) (App. A). Previous work has attempted to improve the learning of artificial agents using gain modulation, such as the Stroop model (Cohen et al., 1990; 2004) or attention mechanisms (Lindsay et al., 2020). Another prominent approach to neuromodulators and cognitive control is Kenji Doja's theory of neuromodulation as a meta-learning mechanism (Doya, 2002), where each neuromodulator is assigned to a specific function in a reinforcement learning setting. Dopamine (Westbrook & Braver, 2016) is proposed to be the error signal in reward prediction (perhaps tasks engagement modulation $\psi_\tau(t)$), serotonin is the time scale of reward prediction (the discount factor $\gamma$ in our setting), noradrenaline (Cohen et al., 1990; Aston-Jones & Cohen, 2005; Shenhav et al., 2013) controls randomness in action selection (also believed to activate different neural paths as in (Cohen et al., 2004), as our gain modulation setting), and acetylcholine (Yu & Dayan, 2005; Ren et al., 2022) controls the speed of updates (as learning rate, which we optimized in Section 4 and Appendix F.2). In summary, **we believe our framework could be tested against neural recordings to validate normative theories of neuromodulators such as the one described here (Doya, 2002; Shenhav et al., 2013)**.

All our work as been simulated in linear networks. **We provide a first approach to non-linear network control** (App. I) by approximating the gradient flow of a non-linear network using first-order Taylor expansion around the mean of the data distribution, then maximizing reward using the gain modulation model. Since the network dynamics of the non-linear network are approximately linear for small weights (and $\tanh(\cdot)$ non-linear activation function), the control obtained when optimizing expected return still speeds up learning in the non-linear network. Given the necessary equations from the learning model, further analysis can be done on, for example, linear recurrent networks, which can be used for complex decoding depending on the properties of the recurrent connections (Bondanelli & Ostojic, 2020). Closed form non-linear dynamics approaches such as the teacher-student settings (Goldt et al., 2019; Ye & Bors, 2022), and mean-field theory (Mignacco et al., 2020; Bordelon & Pehlevan, 2022; Bordelon et al., 2023) are promising direction to extend our framework to non-linear networks and reinforcement learning dynamics.

## B.1 PURPOSE OF THE CONTROL COST

Adding a control cost term to the optimization is standard in control theory literature to describe, for example, energy consumption or depletion of some resource when applying control, and in general, this cost is minimized. In our work, the control cost serves other purposes as well.

First, it limits the space of control signals to avoid trivial solutions. For example, taking $C = 0$ when optimizing the learning rate in the single neuron case, gives the trivial solution of choosing such that the weight reaches the solution after one step. In the case of MAML, $C = 0$ is required to show the equivalence to our framework (See Section 4 and Appendix F.1). Another case of $C = 0$ giving useful control signals is in the gain modulation case. Intuitively, because the gain modulation speeds up learning, the optimal thing to do would be to use an extremely large gain to arrive at the solution weights fast, but the value of the control signal also changes the prediction in $Y = f(X; w(t), g(t))$ potentially increasing the loss. We ran simulations to verify this (not included) in the same setting as the one in effort allocation in Section 6, except that $C = 0$ and with no restrictions on the size of $G$. The resultant gain modulation is qualitatively similar to the case when $C \neq 0$ but is much

more concentrated and less smooth (figure not included). In this case, the cost is not beneficial to the performance, but our goal is to study the nature of these solutions under different cost assumptions, for example, as in the task engagement case in Section 6, where the cost function has a big impact in the optimal control signal.

The second function of the cost is to describe mental effort when doing cognitively demanding tasks. The control cost introduced in our framework is meant to account for this limited attention or sense of fatigue. We are not assuming a meaning of the control cost yet, but some theories of the effort feeling are metabolic resource depletion (which is a quite controversial hypothesis (Hagger et al. 2016; Randles et al. 2017), the opportunity cost of performing another task in the environment (Agrawal et al., 2022), reflecting a bottleneck on information processing in the brain (Musslick et al., 2020), or computation time in presence of uncertainty (Gershman & Burke, 2023). Links to these theories can be directly tested using our framework.

## C  CONTROL GRADIENT

Given the set of equations given in the general setting in Section 2, here we provide a way to compute the gradient $dV/dg(t)$ in equation 3 to estimate the optimal control signal. We first estimate the value function in equation 2 using a Riemann sum to approximate the integral as

$$V \approx \sum_{i=0}^{N} \delta t \gamma^{t_i} \left[ \eta \mathcal{P}(t_i) - C(g(t_i)) \right] \tag{8}$$

where $\delta t$ is a small time constant (in practice equal to the learning rate of the weights SGD trained in the deep linear network), $t_i$ are the bin values for the time span ($t_0 = 0$, $t_N = T$, and $t_{i+1} = t_i + \delta t$). The performance in our case is a function of the loss $\langle \mathcal{L}(t) \rangle$, which depends on the parameters at each time step $w(t)$ and the control signal $g(t)$ from the input-output in equation 1. The parameters themselves evolve depending on the control signal according to the learning dynamics in equation 1, discretizing this differential equation we can write

$$w(t_i) = w(t_{i-1}) + \delta t \frac{dw(t_{i-1})}{dt} \tag{9}$$

where the last term in rhs depends on the control signal $g(t_i)$ as well. Given these equations, we can explicitly write the dependencies of the parameters and loss function as

$$w(t_i) = w(g(t_{i-1}), g(t_{i-2}), ..., g(t_0), w(t_0)) \tag{10}$$
$$\langle \mathcal{L}(t_i) \rangle = \mathcal{L}(w(t_i), g(t_i)). \tag{11}$$

Making these dependencies explicit in equation 8 and replacing $\mathcal{P}(t_i) = -\mathcal{L}(w(t_i), g(t_i))$, we compute the gradient of the approximated integral with respect to $g(t_j)$ giving

$$\frac{dV}{dg(t_j)} = \sum_{i=1}^{N} \delta t \gamma^{t_i} \frac{d}{dg(t_j)} \left[ -\eta \mathcal{L}(w(t_i), g(t_i)) - C(g(t_i)) \right], \tag{12}$$

$$= \underbrace{-\delta t \gamma^{t_j} \left[ \eta \frac{\partial \mathcal{L}(w(t_j), g(t_j))}{\partial g(t_j)} + \frac{\partial C(g(t_j))}{\partial g(t_j)} \right]}_{\text{instant variation}} - \underbrace{\sum_{i=j+1}^{N} \delta t \gamma^{t_i} \eta \frac{\partial \mathcal{L}(w(t_i), g(t_i))}{\partial w(t_i)} \frac{\partial w(t_i)}{\partial g(t_j)}}_{\text{future variation}}. \tag{13}$$

Finally, the optimal control signal can be found by computing

$$g_{k+1}(t_i) = g_k(t_i) + \alpha_g \frac{dV}{dg_k(t_i)} \tag{14}$$

for every $i = 0, 1, ...N$, where $k$ is the iteration index. Note that, the optimal $g^*(t_j)$ such that $dV/dg(t_j) = 0$ will depends on every other $g^*(t_{i \neq j})$. Because the entire control signal $g(t)$ is planned at time step $t = 0$, then for $\gamma = 0$ the integral only considers the term $v(0)dt$ leading to no control. This is different from what is expected from an agent in common settings of reainforcement learning, where the agent makes a decision at each time step, and $\gamma = 0$ is equivalent to maximizing

each $v(t)$ independently. In our setting, $\gamma > 0$ small is virtually equivalent to maximizing instant net reward. From equation 13 it is direct that $\gamma \to 0$ makes the sum (future variations $t_i$ with $i \geq j + 1$) vanish, therefore leaving the gradient only as a function of the loss and control cost at time $t_j$ (meaning maximizing instant reward rate $v(t)$), however, the learning dynamics of $w$ is still depending on $g$, therefore the optimal $g^*(t_j)$ will depend on $g^*(t_i)$ with $i < j$, which can be solved using dynamic programming (Bertsekas, 2012).

## D  LEARNING EFFORT ALGORITHM

The following algorithm is the one implemented to optimize the learning effort or control signal $g(t)$. Time was discretized following Appendix C.

---

**Algorithm 1** Learning Effort Optimization

---

**Input:** A learning system (a input-output mapping equation, and a learning dynamics equation as in equation 1), a task $\mathcal{T}$, learning period $T$, initialize $g_0(t_i)$ for every $i = 0, ..., N$ (with $t_0 = 0$ and $t_N = T$), reward conversion $\eta$, control cost $C(g(t))$, parameters $w(t_0)$, number of gradient updates on the control signal $N_k$, control learning rate $\alpha_g$.
**for** $k = 0$ **to** $N_k$ **do**
    set $V = 0$
    **for** $i = 0$ **to** $N$ **do**
        Compute $w(t_i)$ for every $i$ using the parameters updates in equations 9.
        Compute $Y_i = f(X; w(t_i), g_k(t_i))$
        Compute $\mathcal{P}(t_i)$, and $R(t_i) = \eta \mathcal{P}(t_i)$ (e.g $\mathcal{P}(t_i) = -\langle \mathcal{L}(t_i) \rangle_{XY}$)
        Compute $v(t_i) = R(t_i) - C(g_k(t_i))$
        $V \leftarrow V + v(t_i) \cdot \delta t$
    **end for**
    **for** $i = 0$ **to** $N$ **do**
        $g_{k+1}(t_i) = g_k(t_i) + \alpha_g \frac{dV}{dg_k(t_i)}$
    **end for**
**end for**
**Output:** Optimized control signal $g_{N_k}$.

---

## E  RELATION TO THE EXPECTED VALUE OF CONTROL (EVC) THEORY

In EVC theory (Shenhav et al., 2013), the authors proposed a model that accounts for cognitive control allocation by estimating the action-value function (or signal-value in this case) for each possible control signal, making explicit the cost of taking an action. It also suggests that the dorsal anterior cingulate cortex (dACC) is the part of the brain responsible for integration and computation of all of most quantities needed in the EVC theory, which are the expected payoff of a controlled process, amount of control needed and the cost associated to executing control.

The EVC is posed in a reinforcement learning setting where the control cost is made explicit, and the EVC quantity is the action-value (Q-value function). Let's start from the bellman equation for $q_\pi$ as

$$q_\pi(s, a) = \sum_{s', r} p(s', r | s, a) \left[ \bar{r} + \gamma \sum_{a'} \pi(a' | s') q_\pi(s', a') \right], \tag{15}$$

where $s$ denotes the current state, $a$ the action taken $\bar{r} = r - C(a)$ with $C(a)$ the control cost ($a$ being the control signal), $r$ the reward received from the environment (see equation 1 and 2 in (Shenhav

et al., 2013)), and $\pi$ a given policy (which will end up being our control signal $g(t)$ later on), then

$$q_\pi(s, a) = \sum_{s', r} p(s', r | s, a) \left[ r - C(a) + \gamma \sum_{a'} \pi(a' | s') q_\pi(s', a') \right],$$
(16)

$$EVC(\text{state} = s, \text{signal} = a) = q_\pi(s, a) = \sum_{s', r} p(s', r | s, a) \left[ r + \gamma \sum_{a'} \pi(a' | s') q_\pi(s', a') \right] - C(a).$$
(17)

The expected value of control EVC is the action-value function when writing the cost of taking actions (control) is explicit. We now show that quantity is equivalent to the cumulative reward shown in equation 2 in the main text under certain conditions. We first index the state $s$ with time, taking a non-stochastic policy of the control signal at each state $a = g(t)$ for that particular state, from one step to another there is a small change in time $\delta t$ (making state transition deterministic, $p(s' = t + \delta t | s = t) = 1$), we also re-define reward and cost as a reward/time units, giving

$$EVC(s = t, a = g(t)) = \sum_r p(r | t, g(t)) \left[ r\delta t + \gamma^{\delta t} EVC(s = t + \delta t, a = g(t + \delta t)) \right] - C(g(t))\delta t$$
(18)

$$= \delta t \left( \mathbb{E}\left[ r | t, g(t) \right] - C(g(t)) \right) + \gamma^{\delta t} EVC(s = t + \delta t, a = g(t + \delta t))$$
(19)

$$= \delta t \left( \mathbb{E}\left[ r | t, g(t) \right] - C(g(t)) \right) + \gamma^{\delta t} \delta t \left( \mathbb{E}\left[ r | t + \delta t, g(t + \delta t) - C(g(t + \delta t)) \right] \right)$$
$$+ \gamma^{2\delta t} EVC(s = t + 2\delta t, a = g(t + 2\delta t)).$$
(20)

We can keep unrolling the $EVC$ term in the previous equation until the termination of the task at time $T$, where $r = 0$ for any $t > T$, indexing time as $t_0 = 0, t_N = T$, and $t_{i+1} = t_i + \delta t$

$$EVC = \sum_{i=0}^{N} \delta t \gamma^{t_i} \left[ \mathbb{E}\left[ r | t, g(t) \right] - C(g(t_i)) \right]$$
(21)

which is equation 8. Then, taking the limit $\delta t \to 0$ we recover the integral form in equation 2. For further discussion about the cognitive neuroscience literature see App. B.

## F    RELATION TO META-LEARNING ALGORITHMS IN MACHINE LEARNING

Our optimization framework is related to other meta-learning algorithms in the machine learning literature. Here we provide a formal description of the relation between our framwork and two well-established meta-learning algorithms, *Model-Agnostic Meta-learning for Fast Adaptation of Deep Networks* (MAML (Finn et al., 2017)) and *Bilevel Programming for Hyperparameter Optimization and Meta-Learning* (Franceschi et al., 2018), as well as simulations details of the results shown in Figure 3 in Section 4.

We highlight that there are scalable meta-learning algorithm methods in the literature (Rajeswaran et al., 2019; Deleu et al., 2022) which are able to meta-learn variables in state-of-the-art architectures. However, these methods rely on simplifying the meta objective, restricting the meta-variables (making them smaller for tractability), or using extra iterative processes to approximate gradients. This is fundamentally different from what we are able to achieve in our framework. In most experiments of our paper, each step in our outer loop considers the entire inner loop learning trajectory and computes the gradient of the meta-loss at all time steps to capture the effect of the meta-parameters across the entire learning trajectory, not just the last step as in the referenced work. In addition, our meta-parameters can be as complex as the (size of the network) times (inner loop training iterations), in other words, our meta-variables also depend on time, increasing the complexity of the optimization problem. We are solving the full complex meta-learning problem (which is the desired target in both references), by considering a simpler model, instead of approximating our computation. This will provide insight on the *ideal* meta-objective in complex non-linear learning dynamics which is intractable in large state-of-the-art learning architectures.

### F.1 MODEL AGNOSTIC META-LEARNING

The formal description of MAML is as follows. Consider a set of tasks $\mathcal{T}_i$, and a function $f_\theta$ parametrized by $\theta$. For each $\mathcal{T}_i$, there is a loss function specific for task $i$ denoted as $\mathcal{L}_{\tau_i}(f_\theta)$. They denote the parameters after one gradient step following the loss on one specific task as

$$\theta_i' = \theta - \alpha \nabla_\theta \mathcal{L}_{\tau_i}(f_\theta) \tag{22}$$

with $\alpha$ being the learning rate of the inner loop. Then, the meta-objective is

$$\min_\theta \sum_{\mathcal{T}_i \sim p(\mathcal{T})} \mathcal{L}_{\mathcal{T}_i}\left(f_{\theta_i'}\right) = \min_\theta \mathcal{L}_{\text{MAML}}, \tag{23}$$

which is minimized by stochastic gradient descent on the model parameters

$$\theta = \theta - \alpha_M \nabla_\theta \sum_{\mathcal{T}_i \sim p(\mathcal{T})} \mathcal{L}_{\tau_i}(f_{\theta_i'}), \tag{24}$$

with $\alpha_M$ being the learning rate of the outer loop or meta-iteration. This optimization is minimizing the loss function *of next update steps* across all available tasks, leading to a set of parameters $\theta$ that can rapidly be adapted to solve a specific task within the task distribution. Now, we adjust the parameters of our framework to fit the description of MAML. First, we set our control signal $g = \theta$, which in the two-layer linear network corresponds to the initial weights $g = \theta = (W_1(t = 0), W_2(t = 0))$. Then, we define the performance of the network as

$$\mathcal{P}(t_i) = - \sum_{\mathcal{T}_i \sim p(\mathcal{T})} \langle \mathcal{L}_{\mathcal{T}_i}(t_i) \rangle \tag{25}$$

where $\langle \mathcal{L}_{\mathcal{T}_i}(t_i) \rangle$ is the loss function at time $t_i$ after just training the initial parameters under the task $\mathcal{T}_i$. Then we can write our cumulative reward as

$$V \approx \sum_{i=1}^{N} \delta t \gamma^{t_i} \left[ \eta \mathcal{P}(t_i) - C(g) \right]. \tag{26}$$

Making $\eta = 1$, $\gamma = 1$, $C = 0$ for any $g$, we recover Multi-Step MAML (Ji et al., 2020), and taking $N = 1$ (this is considering one step) we recover standard MAML optimization

$$\max_g V = -\mathcal{P}(t_1) = - \sum_{\mathcal{T}_i \sim p(\mathcal{T})} \langle \mathcal{L}_{\mathcal{T}_i}(t_1) \rangle = -\mathcal{L}_{\text{MAML}} \tag{27}$$

where at $t_1$ we are considering one update step only from the initial parameters $g = (W_1(t = 0), W_2(t = 0))$. Finally, maximizing the value in the previous equation using algorithm D is equivalent to optimizing standard MAML, considering more time-steps as in equation 26 we recover multi-step MAML. The simulation results are shown in Figure 6 and 7, and works as follows: We created a set of tasks $\mathcal{T}_i$ with pairs of MNIST numbers, (0, 1), (7, 1), (8, 9), (3, 8) and (5, 3) (see App. J). Then we picked a range of time-steps to consider during the optimization of the initial weights $g$ (control signal) made with a from a `linspace` starting from 1 to 340 (including) every 20 steps (except between 1 to 20 where there is a difference of 19), we call these *Optimized steps* as in Figure 6 and 7. We evaluate how good is the loss dynamics starting from the optimized initial conditions $g$ throughout 8000 updates steps on each task as shown in Figure 6 and 7 as *eval steps*. We obtain the cumulative loss eval steps in Figure 7 by integrating these dynamics throughout the evaluation time after optimizing initial conditions.

In Figure 6, it is shown that considering more time-steps in the optimization (*Optimized steps*) is beneficial for the resulting dynamics. The more steps are considered, the faster the learning after optimizing for initial parameters. Something important to note is that there is a qualitative difference in the optimization when considering 1 steps, vs considering multiple steps. As mentioned in Section 4 in the main text, considering only one step ahead is a myopic optimization initial condition. Immediately after considering a few steps ahead, the loss dynamics in the evaluation steps is improve very quickly (Figure 7), and the one step ahead loss (the one considered when optimizing only one step) is reduced even more after considering a few more optimized steps, as shown in Figure 6l, presumably because the gradient over the initial parameters can see the dynamics after one step ahead,

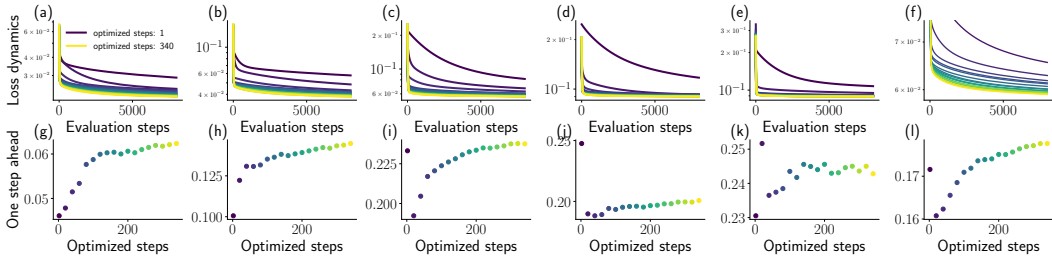

Figure 6: Simulating multi-step MAML with the *learning effort* framework: The first row is the loss dynamics evaluated through 8000 updates steps, starting from the optimized parameters found by MAML, the color code shows the number of optimized steps considered in the meta-objective (1 step is standard MAML). The second row shows the one step ahead when considering different number of optimized steps (same color code), in other words, it is the first loss value from the curves in the first row. From **(a)** to **(e)** columns, the results for the binary regression tasks for MNIST pairs (0, 1), (7, 1), (8, 9), (3, 8) and (5, 3) is shown respectively, with the last column **(f)** being the average across tasks $\mathcal{L}_{MAML}$.

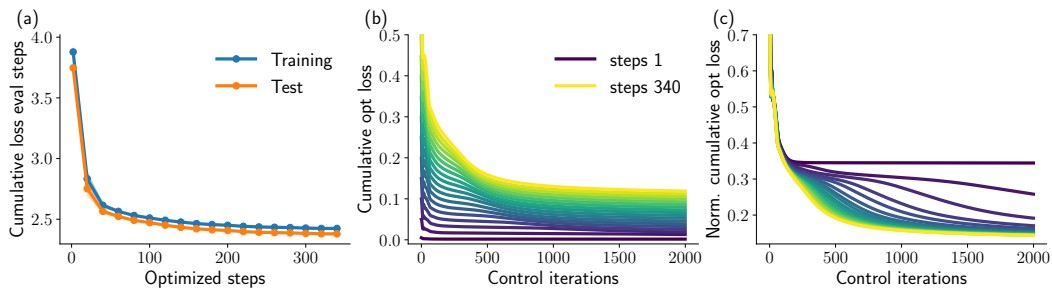

Figure 7: Optimization results on Multi-step MAML. **(a)**: Cumulative loss eval steps (integral of curves in Figure 6) evaluated on training and test sets in MNIST pairs. **(b)**: Actual loss considered during the optimization vs control iterations (finding the best initial conditions $W_1(0)$ and $W_2(0)$). **(c)**: Same as **(b)** but normalized by its maximum for visualization purposes.

perhaps finding better solutions through looking at more steps ahead. Then, there is a transition, where after considering around 180 steps ahead (Figure 6l), the one-step ahead loss increases while improving the loss dynamics in the evaluation steps even more, sacrificing immediate loss for a better overall cumulative dynamics. This hypothesis is supported by looking at Figure 7c where optimizing one step ahead converges in a few iterations over the initial parameters $g$. In contrast, when considering more steps, the optimization goes beyond this plateau and the speed at which this plateau is skipped increases with the number of optimized steps considered. The myopic solution is not able to optimize further since it doesn't have information of the learning dynamics, which is provided when considering multiple steps as shown by these results. In Table 2 we summarize the parameters used for the simulation.

## F.2 BILEVEL PROGRAMMING

In Bilevel Programming (Franceschi et al., 2018) which unifies gradient-based hyperparameter optimization and meta-learning algorithms. They show an approximated bilevel programming method and the conditions to guarantee convergence to the exact problem. The setting described in this work show meta-learning problems as an inner and outer objective $L_\lambda$ and $E(w, \lambda)$ respectively, where $w$ are the parameters of the model and $\lambda$ are the hyperparameters (equations 3 and 4 in (Franceschi et al., 2018)), and the approximated problem corresponds to doing a few update steps only in the inner objective, meaning the loss of the inner loop is not fully minimized. The first

difference between our setting and Bilevel Programming is the method they use to optimize the meta-parameters called reverse-hypergradient method (Franceschi et al., 2017). Another difference, is that we extend the bilevel optimization setting to have a normative meaning through the inclusion of a control cost, discount factor and we further simplify it by using the average learning dynamics obtained from using the gradient flow limit in the two-layer linear network. As an example of this, we find the optimal learning rate $\alpha(t)$ throughout learning that maximizes the value function for the semantic task, to give more intuition on the impact of learning rate changes for complex step-like learning curves. In our implementation, we find the optimal learning rate using a surrogate variable $\rho(t)$ that facilitates the optimization. We use the learning effort framework to train a two layer linear network on the semantic dataset, just by modifying the learning dynamics as

$$\tau_w \frac{dW_1}{dt} = (1 + \rho(t)) \left[ W_2^T \left( \Sigma_{xy}^T - W_2 W_1 \Sigma_x \right) - \lambda W_1 \right],$$ (28)

$$\tau_w \frac{dW_2}{dt} = (1 + \rho(t)) \left[ \left( \Sigma_{xy}^T - W_2 W_1 \Sigma_x \right) W_1^T - \lambda W_2 \right].$$ (29)

We define $\rho(t) = g(t)$ as our control signal (effort signal), therefore with $\rho = 0$ we recover the baseline learning dynamics of a network trained with SGD, and the effective learning rate is $\alpha(t) = (1 + \rho(t))\alpha$ with $\alpha$ the baseline learning rate. We set $\eta = 1$ and the cost $C(\rho(t)) = \beta \left( \rho - \text{offset} \right)^2$. The optimal learning rates $\alpha(t)$ and average training loss resulting from the control signal are depicted in Figure 8.

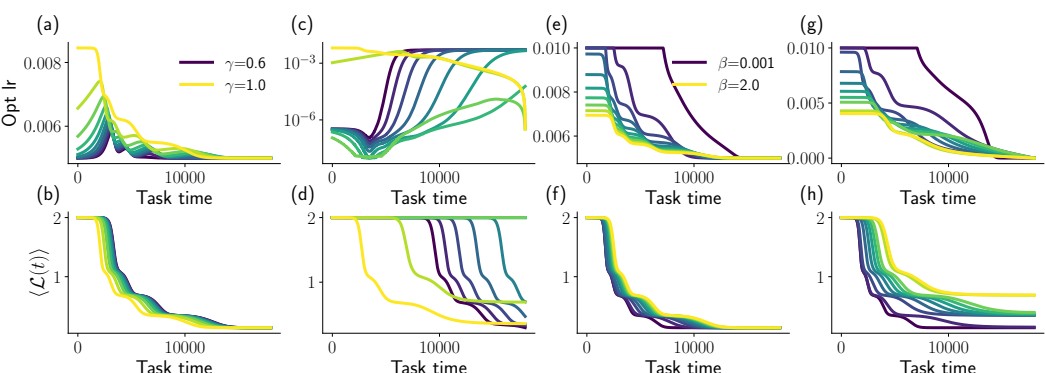

Figure 8: Learning rate optimization. Top row: Optimal learning rates $\alpha(t)$. Bottom row: Resulting average dynamics. **(a)**, **(b)**, **(c)** and **(d)**: results when varying $\gamma$, **(a)** and **(b)** when the offset = 0 (baseline dynamics has 0 cost), **(c)** and **(d)** when the offset= -1 (any dynamics is costly). **(e)**, **(f)**, **(g)** and **(h)**: Results for varying $\beta$.

As mentioned in Section 4 in the main text, the control signal as the learning rate presents qualitatively the same behavior as in the single neuron case as shown in FIgure 8. More control is allocated when $\gamma$ increases due to pay off in the future, and more control is used when $\beta$ is decreased. The step-like shape in these plots is from learning each step in the hierarchy of the semantic task. Something important to notice is the results in the last column of Figure 8, where depending on the cost of using control, the optimal solution is to learn some, but not all of the levels of the hierarchy. A higher cost of control leads to fewer levels learned in the hierarchy, meaning that deeper and harder levels in the structure might not be worth it if it is too costly for learning effort. The parameters used for this simulation are shown in Table 3.

## F.3 META-LEARNING THE LEARNING RULE

Another way to use our framework is to find optimal learning rules that maximize value. To show this, we trained a neural network where the learning rules are parametrized, and we optimized these parameters as being the control signal in our framework. We implemented a model by Cao et al. (2020), where the plasticity rule is a function of two parameters $\gamma$ and $\eta$ (which are not the same as

the quantities defined in Sec. 2 for our framework), giving update rules of the form:

$$\Delta W_1 = \Delta W_1^C(\gamma) + \Delta W_1^H(\eta), \quad \Delta W_2 = \Delta W_2^C(\gamma) \tag{30}$$

where $W_i^C$ denotes a contrastive Hebbian plasticity update, and $W_i^H$ denotes a Hebbian update, which are controlled by $\gamma$ and $\eta$ respectively. These rules are described by

$$\Delta W_1^C(\gamma) = \alpha W_2^T(y - \hat{y})x^T \tag{31}$$

$$\Delta W_2^C(\gamma) = (y - \hat{y})x^T W_1^T + \gamma(yy^T - \hat{y}\hat{y}^T)W_2 \tag{32}$$

$$\tag{33}$$

for the contrastive Hebbian, with $\alpha$ being the learning rate, and

$$\Delta W_1^H(\eta)_{(i,:)}^T = \begin{cases} \eta(W_1 x)_{(i,:)} \left( x^T - (W_1 x)_{(i,:)} W_1(i,:)^T \right), & \text{if } \eta > 0 \\ \eta(W_1 x)_{(i,:)}/(1 + \|W_1(i,:)^T\|_2^2), & \text{otherwise} \end{cases} \tag{34}$$

where $(i,:)$ denotes row $i$. Depending on the values of $\gamma$ and $\eta$, the learning rules changes as depicted in Figure 9c, spanning Contrastive Hebbian learning, gradient descent, Quasi-predictive coding, Hebbian and Anti-Hebbian learning (see Cao et al. (2020) for more details). Assuming a linear network, and taking the gradient flow limit (as done in all of the other settings), we can write the average learning dynamics equations as

$$\left\langle \Delta W_1^C(\gamma) \right\rangle_{x,y} = \alpha W_2^T \left( \Sigma_{xy}^T - W_2 W_1 \Sigma_x \right) \tag{35}$$

$$\left\langle \Delta W_2^C(\gamma) \right\rangle_{x,y} = \alpha \left( \Sigma_{xy}^T - W_2 W_1 \Sigma_x \right) W_1^T + \alpha\gamma(\Sigma_y W_1^T W_2^T - W_2 W_1 \Sigma_x W_1^T W_2^T)W_2 \tag{36}$$

and

$$\left\langle \Delta W_1^H(\eta)_{(i,:)}^T \right\rangle_{x,y} = \begin{cases} \eta \left[ W_1 \Sigma_x - \left( W_1^2 diag(\Sigma_x) \right) \circ W_1 \right]_{(i,:)}, & \text{if } \eta > 0 \\ \eta(W_1 \Sigma_x)_{(i,:)}/(1 + \|W_1(i,:)^T\|_2^2), & \text{otherwise} \end{cases}. \tag{37}$$

We can define $\eta(t)$ and $\gamma(t)$ as the control signals, and find the optimal learning rules through time using our framework by updating the control signal by

$$\eta^{k+1}(t_i) = \eta^k(t_i) + \alpha_\eta \frac{dV}{d\eta(t_i)}, \quad \gamma^{k+1}(t_i) = \gamma^k(t_i) + \alpha_\eta \frac{dV}{d\gamma(t_i)}, \tag{38}$$

using algorithm 1. The results of this simulation are simulated below in Figure 9, and the hyperparameters of the model are the same as in Cao et al. (2020). Figure 9a shows the improvement in dynamics after optimizing the learning rules, and the improvement in minimizing the negative value function $-V$ in Figure 9b. We initialized $\eta(t) = 0$ and $\gamma(t) = 0$ which is equivalent to stochastic gradient descent. By updating these control signals using equation 38, we obtained a mix between contrastive Hebbian and standard Hebbian rules (blue dot in Figure 9c), which is consistent with the results found by Cao et al. (2020). We bounded the solution to $-0.2 < \eta < 0.2$ and $-1 < \gamma < 1$, so the optimal control is a fixed value in the upper-right corner as shown in Figure 9. No cost was imposed on the control signal.

### F.4 META-LEARNING BEYOND CONTROL

Our framework is based on the fact that we have at least one free parameter that does not evolve according to a given learning rule, described by the learning dynamic in equation 7. To the best of our knowledge, we think we can adapt our framework to learn any free parameters not governed by the learning dynamics, such as control signals or hyperparameters to maximize value. Still, there is an important part of the literature on meta-learning, where no parameters are explicitly trained to learn an outer loop loss or meta-learning goal, but rather, it emerges from training on a large task distribution or a large number of tasks (Wang et al., 2017; Team et al., 2021). In these cases, there is no control signal or set of parameters that are explicitly optimized in an outer loop, all parameters are trained in all of the tasks, giving rise to a solution that can perform one-shot learning in unseen tasks. The reasoning for this is that, in order to solve a distribution of tasks with the same set of parameters available to the agent, the agent must find strategies that work for all tasks simultaneously, therefore giving rise to "meta-solutions" that can generalize well, or that can be learned within a few

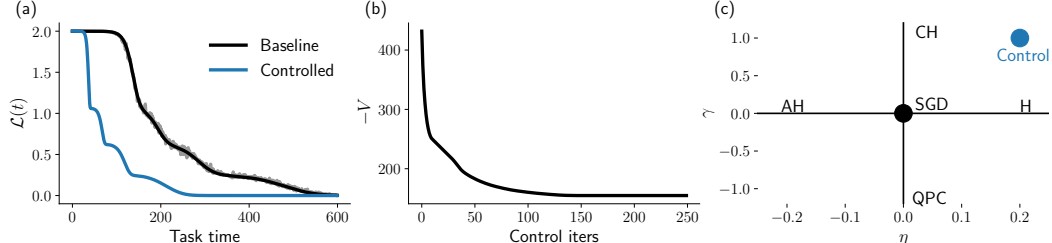

Figure 9: Optimal learning rule optimization. (a): Learning dynamics comparing baseline ($\gamma(t) = 0$ and $\eta(t) = 0$) vs optimized. (b) Improvement in maximizing $V$ (minimizing $-V$). (c) The optimal regime for learning rules (blue dot) that maximize value $V$. CH: Contrastive Hebbian, SGD: Gradient descent, QPC: Quasi-predictive coding, H: Hebbian, AH: Anti Hebbian.

trials. Another set of meta-learning algorithms is "memory-based" meta-learning (Ritter et al., 2018; Genewein et al., 2023). These models use stored memory of past experiences to perform tasks (see "Neural Episodic Control", Pritzel et al. (2017)). The meta-learning part also emerges from training by using memory, and it is not explicitly controlled as in our case. Also, learning dynamics for neural networks learning action-value, or value-functions are not easy to describe (Bordelon et al., 2023) as in the regression problem we are solving in our paper. We could adapt our framework to account for these types of models in the future. Our framework is likely to be most insightful in cases where a subset of parameters are optimized toward a distinct meta-learning objective in an outerloop.

# G  LEARNING DYNAMICS SOLUTIONS

## G.1  SINGLE NEURON MODEL

Here we provide the learning dynamics solution for the single-neuron model given a learning effort signal $G(t)$. The input-output mapping of this network is

$$\hat{y} = xw(t) \cdot (1 + g(t)) \tag{39}$$
$$= xw(t)\tilde{g}(t) \tag{40}$$

with $\tilde{g}(t) = (1 + g(t))$, and the input samples $x_i$ carry information of its respective label $y_i$ as $x_i \sim \mathcal{N}(y_i\mu_x, \sigma_x^2)$, with $y_i = (1 - 2\xi)$ and $\xi \sim \text{Bernoulli}(1/2)$, $g(t)$ is our learning effort signal ($g_{\max} \geq g(t) \geq 0$), and $w(t)$ is the neuron weight. Using MSE loss and $L_2$ regularization over the weights, the complete loss function for this problem is

$$\mathcal{L} = \frac{1}{2}\left[y - \hat{y}\right]^2 + \frac{\lambda}{2}w^2. \tag{41}$$

Learning the weight by gradient descent, by taking the derivative of the loss $\mathcal{L}$ with respect to the weights, we get

$$\frac{\partial \mathcal{L}}{\partial w} = -(y - \hat{y})\frac{\partial \hat{y}}{\partial w} + \lambda w \tag{42}$$
$$= x^2 w\tilde{g}^2 - yx\tilde{g} + \lambda w. \tag{43}$$

Updating the weights using gradients steps (gradient step iteration index $k$, and sample index $b$ from a batch with size $B$) gives

$$w(t_{k+1}) = w(t_{k+1}) - \alpha\frac{1}{B}\sum_{b=1}^{B}\frac{\partial \mathcal{L}(y_b, x_b)}{\partial w}. \tag{44}$$

Taking the gradient flow limit $\alpha \to 0$, the number of samples given to the model per unit of time goes to infinity, converting the average to an expectation over samples (Saxe et al., 2019; Elkabetz &

Cohen, 2021), we obtain

$$\tau_w \frac{dw}{dt} = -\left\langle \frac{\partial \mathcal{L}}{\partial w} \right\rangle \tag{45}$$

$$= \mu \tilde{g}(t) - w(t) \left( \langle x^2 \rangle \tilde{g}^2(t) + \lambda \right). \tag{46}$$

here introducing $\tau_w$ to control the time scale of the weight evolution. We can solve this differential equation for $w(t)$ for any unknown $g(t)$ using an integrating factor

$$M(t) = \exp \left( \int_0^t \frac{\langle x^2 \rangle \tilde{g}^2(t') + \lambda}{\tau_w} dt' \right). \tag{47}$$

Multiplying both sides of equation 46

$$M(t) \frac{dw}{dt} + M(t)' w = M(t) \frac{\mu \tilde{g}}{\tau_w} \tag{48}$$

$$\frac{d}{dt} \left( M(t) w(t) \right) = M(t) \frac{\mu \tilde{g}}{\tau_w}. \tag{49}$$

Integrating both sides and solving for $w(t)$ we get

$$w(t) = M^{-1}(t) \left[ \int_0^t dt' M(t') \frac{\mu \tilde{g}(t')}{\tau_w} + w(0) \right]. \tag{50}$$

## G.2 SINGLE LAYER NETWORK

Here we provide the learning dynamics solution for a single-layer linear network given a learning effort signal $G(t)$. The input-output mapping of this network is

$$\hat{Y}_i(t) = ((\mathbb{1} + G(t)) \circ W(t)) X_i, \tag{51}$$

$$= (\tilde{G}(t) \circ W(t)) X_i. \tag{52}$$

where $X_i \in \mathbb{R}^I$ ($i$ being sample index), $W(t) \in \mathbb{R}^{O \times I}$, $\tilde{G}(t) = G(t) + \mathbb{1}$, $\mathbb{1}, G(t) \in \mathbb{R}^{O \times I}$ ($\mathbb{1}$ is a matrix with ones) and $\circ$ denotes element-wise multiplication. Using mean square error as the loss function, with $L_2$ regularization on the weights

$$\mathcal{L}_i(t) = \frac{1}{2} \|Y_i - \hat{Y}_i(t)\|^2 + \frac{\lambda}{2} \|W(t)\|_F^2, \tag{53}$$

with $Y_i$ being the target output of sample $i$. Taking the gradient flow limit of the batch updates as in equations 44 and 45 we get

$$\tau_w \frac{dW}{dt} = -\left\langle \frac{\partial \mathcal{L}}{\partial W} \right\rangle, \tag{54}$$

$$= \Sigma_{xy}^T \circ \tilde{G}(t) - \left[ (W(t) \circ \tilde{G}(t)) \Sigma_x \right] \circ \tilde{G}(t) - \lambda W(t), \tag{55}$$

with $\Sigma_{xy} = \langle X Y^T \rangle$ and $\Sigma_x = \langle X X^t \rangle$. To solve this equation, we vectorize all the matrices, naming $\text{vec}(\cdot)$ the vectorization operator, $\text{diag}(v) \in \mathbb{R}^{n \times n}$ converts a vector $v \in \mathbb{R}^n$ into a diagonal matrix, using $\text{vec}(ABC) = (C^T \otimes A)\text{vec}(B)$ property, where $\otimes$ is the Kronecker product, then we denote

$$\text{vec}(W) = w, \quad \text{diag}\left( \text{vec}(\tilde{G}) \right) = \tilde{g}, \quad \text{vec}(\Sigma_{xy}^T \circ \tilde{G}) = w_g \tag{56}$$

we can rewrite the differential equation in 55 as

$$\tau_w \frac{dw}{dt} = w_g - \left[ \left( \Sigma_x^T \otimes I \right) \tilde{g}^2 + \lambda I \right] w. \tag{57}$$

Then, solving for $w(t)$ using the integrating factor trick

$$M(t) = \text{expm} \left( \int_0^t \frac{dt'}{\tau_w} \left[ \left( \Sigma_x^T \otimes I \right) \tilde{g}^2 + \lambda I \right] \right), \tag{58}$$

denoting $\text{expm}(\cdot)$ as the matrix exponential function, we obtain the evolution of the weights given any control signal $G(t)$ described as

$$w(t) = M^{-1}(t) \cdot \left( \int_0^t M(t') w_g dt' + w(0) \right). \tag{59}$$

To find the optimal control signal $G(t)$ (multiplication factor on weights) we used algorithm 1 to maximize expected return in equation 2 by iterating

$$G^{k+1}(t) = G^k(t) + \alpha_g \frac{dV}{dG(t)} \tag{60}$$

with $k$ the update iteration index, and $\alpha_g$ the control signal learning rate.

## H  TWO-LAYER LINEAR NETWORK

Taking $\hat{Y} = W_2 W_1 X$, where $X \in \mathbb{R}^I$, $\hat{Y} \in \mathbb{R}^O$, $W_1(t) \in \mathbb{R}^{H \times I}$ and $W_2(t) \in \mathbb{R}^{O \times H}$ are the first and second layer weights (dropping time dependency of the weights to simplify notation), the loss function is

$$\mathcal{L} = \frac{1}{2} \| Y - \hat{Y} \|^2 + \frac{\lambda}{2} \left( \| W_2 \|_F^2 + \| W_1 \|_F^2 \right), \tag{61}$$

$$= \frac{1}{2} \text{Tr} \left( \left( Y - \hat{Y} \right) \left( Y - \hat{Y} \right)^T \right) + \frac{\lambda}{2} \left( \| W_2 \|_F^2 + \| W_1 \|_F^2 \right), \tag{62}$$

$$= \frac{1}{2} \left[ \text{Tr} \left( Y Y^T \right) - 2 \text{Tr} \left( Y X^T W_1^T W_2^T \right) + \text{Tr} \left( W_2 W_1 X X^T W_1^T W_2^T \right) \right] + \frac{\lambda}{2} \left( \| W_2 \|_F^2 + \| W_1 \|_F^2 \right). \tag{63}$$

Taking the derivative of $\mathcal{L}$ with respect to the weights $W_2$ and $W_1$, in general $\frac{\partial}{\partial W} \text{Tr} \left( A W^T B \right) = BA$ and $\frac{\partial}{\partial W} \text{Tr} \left( A W B W^T C \right) = A^T C^T W B^T + C A W B$, then we get

$$\frac{\partial \mathcal{L}}{\partial W_1} = -W_2^T Y X^T + W_2^T W_2 W_1 X X^T + \lambda W_1, \tag{64}$$

$$\frac{\partial \mathcal{L}}{\partial W_2} = -Y X^T W_1^T + W_2 W_1 X X^T W_1^T + \lambda W_2. \tag{65}$$

Updating the weights using gradients steps (layers $i = \{1, 2\}$, gradient step iteration index $k$, and sample index $b$ from a batch with size $B$) gives

$$W_i(t_{k+1}) = W_i(t_{k+1}) - \alpha \frac{1}{B} \sum_{b=1}^{B} \frac{\partial \mathcal{L}(Y_b, X_b)}{\partial W_i}. \tag{66}$$

Taking the gradient flow limit $\alpha \to 0$, number of samples given to the model per unit of time goes to infinity, converting the average to an expectation over samples (Saxe et al., 2019; Elkabetz & Cohen, 2021),

$$\tau_w \frac{dW_i}{dt} = - \left\langle \frac{\partial \mathcal{L}}{\partial W_i} \right\rangle, \tag{67}$$

obtaining

$$\tau_w \frac{dW_1}{dt} = W_2^T \left( \Sigma_{xy}^T - W_2 W_1 \Sigma_x \right) - \lambda W_1, \tag{68}$$

$$\tau_w \frac{dW_2}{dt} = \left( \Sigma_{xy}^T - W_2 W_1 \Sigma_x \right) W_1^T - \lambda W_2. \tag{69}$$

Note that, both equations, the input-output mapping ant the learning dynamics are valid simultaneously, then describing the learning system as forward and backward happening at the same time.

## H.1 GAIN MODULATION

In this model, all weights are multiplied by a gain term, which we will optimize to maximize the expected return. The input-output equation of the neural network with gain modulation is $\hat{Y} = \left(W_2 \circ \tilde{G}_2\right)\left(W_1 \circ \tilde{G}_1\right) X = \tilde{W}_2 \tilde{W}_1 X$ and $\tilde{G}_i = (\mathbb{1}_i + G_i)$, where $\mathbb{1}$ is a matrix of ones, and $\mathbb{1}_i$, $G_i$ having the same shape as $W_i$ (again dropping time dependence for weights and gain for notation simplicity), then the loss is

$$\mathcal{L} = \frac{1}{2}\left[\mathrm{Tr}\left(YY^T\right) - 2\,\mathrm{Tr}\left(YX^T\tilde{W}_1^T\tilde{W}_2^T\right) + \mathrm{Tr}\left(\tilde{W}_2\tilde{W}_1 XX^T\tilde{W}_1^T\tilde{W}_2^T\right)\right] + \frac{1}{2}\left(\|W_2\|_F^2 + \|W_1\|_F^2\right). \tag{70}$$

In general

$$\frac{\partial}{\partial W}\,\mathrm{Tr}\left(A\tilde{W}^T B\right) = (BA) \circ G \tag{71}$$

$$\frac{\partial}{\partial W}\,\mathrm{Tr}\left(A\tilde{W}B\tilde{W}^T C\right) = \left(A^T C^T \tilde{W} B + CA\tilde{W}B^T\right) \circ G. \tag{72}$$

Following the same procedure as in Appendix H, we can derive the learning dynamics equations for the weights when using gain modulation:

$$\tau_w \frac{dW_1}{dt} = \left(\tilde{W}_2^T \Sigma_{xy}^T\right) \circ \tilde{G}_1 - \left(\tilde{W}_2^T \tilde{W}_2 \tilde{W}_1 \Sigma_x\right) \circ \tilde{G}_1 - \lambda W_1,$$

$$\tau_w \frac{dW_2}{dt} = \left(\Sigma_{xy}^T \tilde{W}_1^T\right) \circ \tilde{G}_2 - \left(\tilde{W}_2 \tilde{W}_1 \Sigma_x \tilde{W}_1^T\right) \circ \tilde{G}_2 - \lambda W_2. \tag{73}$$

The $G_2$ and $G_1$ that optimize value can be computed iterating

$$G_i^{k+1}(t_i) = G_i^k(t_i) + \alpha_g \frac{dV}{dG_i(t_i)} \tag{74}$$

using algorithm 1.

### H.1.1 CONTROL BASIS

For a set of weights $W_i() \in \mathbb{R}^{m \times n}$, then $G_i(t) \in \mathbb{R}^{m \times n}$ such that $\tilde{W}_i(t) = W_i(t) \circ (\mathbb{1} + G_i(t))$, we can write the control signal as a projection on a basis

$$G_i(t) = \sum_b \nu_i^b(t) G_i^b. \tag{75}$$

Note that the time dependency of $G_i(t)$ comes from $\nu_i^b$ which is the variable it is optimized, instead of $G_i(t)$ directly.

**Neuron Basis**: Take $b$ indexing a row (or column) of a matrix $\in \mathbb{R}^{m \times n}$, then $G_i^b$ has row (or column) as 1s, and 0 everywhere else. For example, if $b$ indexes the rows of $G_i^b$, then

$$G_i^b = \text{row } b \rightarrow \begin{bmatrix} 0 & 0 & \cdots & 0 \\ \vdots & \vdots & \vdots & \vdots \\ 0 & 0 & \cdots & 0 \\ 1 & 1 & \cdots & 1 \\ 0 & 0 & \cdots & 0 \\ \vdots & \vdots & \vdots & \vdots \\ 0 & 0 & \cdots & 0 \end{bmatrix}. \tag{76}$$

This is called neuron basis since $\nu_i^b(t)$ will end up multiplying all of the weights connecting a specific neuron $b$. For example, in the previous $G_i^b$ where the rows are 1s, using this gain modulation on the second layer $W_2$, means that $\nu_i^b(t)$ will multiply the output $b$ of the layer. Changing this by columns means modulation of the input weights per each hidden unit. In this case, the iterations on the control signal to maximize the expected return in equation 2, following algorithm 1 is

$$\nu_i^{k+1}(t_i) = \nu_i^k(t_i) + \alpha_g \frac{dV}{d\nu_i(t_i)}. \tag{77}$$

This procedure was used in the category assimilation task on MNIST and the Semantic Datasetm but with the specific restriction of using a *neuron basis* for $G_2(t)$ as in equation 75, while keeping $G_1(t) = 0$ (no gain modulation on the first layer). The *neuron basis* on the output neurons allows the control signal to change the gain on specific output neurons, therefore changing the gain of the learning signal from a specific category in a classification task. See the results of this specific model in Appendix K.5.

## H.2 DATASET ENGAGEMENT MODULATION

In the engagement modulation, the auxiliary loss used to derive the learning dynamics equation, in this case, is given by

$$\mathcal{L}_{\text{aux}} = \sum_{\tau=1}^{N_\tau} \psi_\tau(t)\mathcal{L}(\hat{Y}_\tau, Y_\tau) + \frac{\lambda}{2}\left(\|W_2\|_F^2 + \|W_1\|_F^2\right), \tag{78}$$

where $N_\tau$ is the number of available datasets, $\psi_\tau(t)$ are the engagement coefficients for dataset $\tau$, $\hat{Y}_\tau$ and $Y_\tau$ are the predictions and target for dataset $\tau$. The network can simultaneously try to solve all of the dataset at the same time since the inputs and outputs per task are concatenated as $X^T = [X_1^T, ..., X_\tau^T, ..., X_{N_\tau}^T] \in \mathbb{R}^I$ and $Y^T = [Y_1^T, ..., Y_\tau^T, ..., Y_{N_\tau}^T] \in \mathbb{R}^O$. Then, taking the gradient with respect to the weights, and taking the gradient flow limit we obtain equation 6. In this equation, $\Sigma_x = \langle XX^T \rangle$ can be expressed as the statistics of each tasks following

$$\Sigma_x = \begin{bmatrix} \Sigma_1 & \cdots & \langle X_1 \rangle \langle X_\tau \rangle^T & \cdots & \langle X_1 \rangle \langle X_{N_\tau} \rangle^T \\ \vdots & \ddots & & & \\ \langle X_\tau \rangle \langle X_1 \rangle^T & & \Sigma_\tau & & \\ \vdots & & & \ddots & \\ \langle X_{N_\tau} \rangle \langle X_1 \rangle^T & & & & \Sigma_{N_\tau} \end{bmatrix}, \tag{79}$$

with $\Sigma_\tau = \langle X_\tau X_\tau^T \rangle$. Since the target $Y_\tau$ is only correlated to the input $X_\tau$, the input-output correlation matrix $\Sigma_{xy\tau} \in \mathbb{R}^{I \times O}$ is

$$\Sigma_{xy\tau} = \begin{bmatrix} 0 & \ldots 0 \ldots & \langle X_1 \rangle \langle Y_\tau \rangle^T & \ldots 0 \ldots & 0 \\ \vdots & & \vdots & & \vdots \\ 0 & \ldots 0 \ldots & \langle X_\tau Y_\tau \rangle^T & \ldots 0 \ldots & 0 \\ \vdots & & \vdots & & \vdots \\ 0 & \ldots 0 \ldots & \langle X_{N_\tau} \rangle \langle Y_\tau \rangle^T & \ldots 0 \ldots & 0 \end{bmatrix}. \tag{80}$$

$$\underbrace{\phantom{aaaaaaaaaaaaaaaaaaaaaaaaaaaaaaaaaaaaaaaaaaaaaaaa}}_{\text{output size O}}$$

The rows of $W_{2\tau} \in \mathbb{R}^{O \times H}$ are replaced with zeros for outputs not contributing to $\hat{Y}_\tau$. From here, the $\psi_\tau(t)$ that optimize value can be computed iterating

$$\psi_\tau^{k+1}(t_i) = \psi_\tau^k(t_i) + \alpha_g \frac{dV}{d\psi_\tau(t_i)} \tag{81}$$

using algorithm 1.

## H.3 CATEGORY ENGAGEMENT MODULATION

This derivation is similar to the engagement modulation, but the engagement coefficient scales the error coming from each of the categories from a classification problem. The loss function is just the mean square error between the labels and the output of the network. The auxiliary loss used to derive the learning dynamics equations can be written as

$$\mathcal{L}_{\text{aux}} = \frac{1}{2}\sum_{c=1}^C [\phi_c(y_c - \hat{y}_c)]^2 + \frac{\lambda}{2}\left(\|W_2\|_F^2 + \|W_1\|_F^2\right), \tag{82}$$

$$= \frac{1}{2}\left[\text{d}(\phi)(Y - \hat{Y})\right]^2 + \frac{\lambda}{2}\left(\|W_2\|_F^2 + \|W_1\|_F^2\right) \tag{83}$$

with $d(\phi) = \text{diag}(\phi)$ and $\phi = [\phi_1, ..., \phi_c, ..., \phi_C]^T$. Then, deriving the learning dynamics equation by learning the weights using backpropagation (as in Appendix 5) we obtain

$$\tau_w \frac{dW_1}{dt} = W_2^T d(\phi)^2 \Sigma_{xy}^T - W_2^T d(\phi)^2 W_2 W_1 \Sigma_x - \lambda W_1,$$

$$\tau_w \frac{dW_2}{dt} = d(\phi)^2 \Sigma_{xy}^T W_1^T - d(\phi)^2 W_2 W_1 \Sigma_x W_1^T - \lambda W_1. \tag{84}$$

The reason for this slight variation compared to the task engagement model, is because for the category engage case, we assume we do not have access to $\langle X_c X_c^T \rangle$ or $\langle X_c Y_c^T \rangle$ which are class-specific quantities of the dataset. From here, the $\phi_c(t)$ that optimize value can be computed iterating

$$\phi_c^{k+1}(t_i) = \phi_c^k(t_i) + \alpha_g \frac{dV}{d\phi_c(t_i)} \tag{85}$$

using algorithm 1.

**Class proportion experiment**: We trained a neural network (same architecture as in this section), and modifying the proportion of classes through time using the category engagement inferred from the optimization. The number of elements per class $b_c(t_i)$ in a batch of size $B$ used in this experiment is

$$b_c(t_i) = \frac{\phi_c(t_i) B}{C} \tag{86}$$

with $i$ indexing the SGD iteration on training the weights $W_1$ and $W_2$.

## I  NON-LINEAR TWO-LAYER NETWORK

As a first approach to applying this same learning effort framework in non-linear networks, we approximated the dynamics by Taylor expanding the non-linearities around the mean to get equations depending on first and second moment of the data distribution. Consider a neural network of the form $\hat{Y} = W_2 f(W_1 X)$ where $f$ is a non-linear function ($\tanh(\cdot)$ used in simulations). Following the same setting and procedure as in Appendix H, the loss function can be written as

$$\mathcal{L} = \frac{1}{2} \left[ \text{Tr}\left(YY^T\right) - 2\,\text{Tr}\left(Y^T W_2 f(W_1 X)\right) + \text{Tr}\left(f(W_1 X)^T W_2^T W_2 f(W_1 X)\right) \right] + \frac{1}{2}\left(\|W_2\|_F^2 + \|W_1\|_F^2\right). \tag{87}$$

Taking the derivative of $\mathcal{L}$ with respect to $W_i$, updating by gradient descend and taking the gradient flow limit as in equations 66 and 67, we obtain

$$\tau_w \frac{dW_2}{dt} = \left\langle Yf(W_1 X)^T - W_2 f(W_1 X) f(W_1 X)^T \right\rangle_{XY} - \lambda W_2, \tag{88}$$

$$\tau_w \frac{dW_1}{dt} = \left\langle \text{diag}(f') W_2^T Y X^T - (W_2 \text{diag}(f'))^T (W_2 f(W_1 X)) X^T \right\rangle_{XY} - \lambda W_1, \tag{89}$$

with $f' = f'(W_1 X)$ is the element-wise application of the non-linear function derivative on $W_1 X$, and $\text{diag}(f')$ a diagonal matrix with $f'$ in the diagonal. Now we take the Taylor expansion of $f$ around the mean of the data distribution,

$$f(W_1 X) \approx f(W_1 \langle X \rangle) + J(W_1 \langle X \rangle)(X - \langle X \rangle) \quad \text{with} \quad J = W_1 \text{diag}(f'(W_1 \langle X \rangle)). \tag{90}$$

Replacing this expansion in equations 88 and 89, using $f' = f'(W_1 \langle X \rangle)$ then taking the expectation $\langle \cdot \rangle_{XY}$ we obtain

$$\tau_w \frac{dW_2}{dt} \approx \langle Y \rangle f(W_1 X)^T + \left[ \Sigma_{xy}^T + \langle Y \rangle \langle X \rangle^J \right] J^T$$
$$- W_2 \left[ f(W_1 X) f(W_1 X)^T + J\Sigma_x J^T + J \langle X \rangle \langle X \rangle^T J^T \right] - \lambda W_2$$

$$\tau_w \frac{dW_1}{dt} \approx \text{diag}(f') \left[ W_2^T \Sigma_{xy}^T - W_2^T W_2 f(W_1 X) X^T W_2^T W_2 J \Sigma_x W_2^T W_2 J \langle X \rangle \langle X \rangle^T \right] - \lambda W_1. \tag{91}$$

In case of using gain modulation, $\hat{Y} = \tilde{W}_2 f\left(\tilde{W}_1 X\right)$ as in Section H.1. Executing same steps as for the case without control (see Appendix H.1), the approximated learning dynamics for the gain modulation case is

$$\tau_w \frac{dW_2}{dt} \approx \left(\langle Y\rangle f\left(\tilde{W}_1 X\right)^T + \left[\Sigma_{xy}^T + \langle Y\rangle \langle X\rangle^J\right] J^T\right) \circ \tilde{G}_2$$

$$- \left(\tilde{W}_2 \left[f\left(\tilde{W}_1 X\right) f\left(\tilde{W}_1 X\right)^T + J\Sigma_x J^T + J\langle X\rangle \langle X\rangle^T J^T\right]\right) \circ \tilde{G}_2 - \lambda W_2,$$

$$\tau_w \frac{dW_1}{dt} \approx \left(\operatorname{diag}(f')\left[\tilde{W}_2^T \Sigma_{xy}^T - \tilde{W}_2^T \tilde{W}_2 f\left(\tilde{W}_1 X\right) X^T \tilde{W}_2^T \tilde{W}_2 J\Sigma_x \tilde{W}_2^T \tilde{W}_2 J\langle X\rangle \langle X\rangle^T\right]\right) \circ \tilde{G}_1 - \lambda W_1,$$
(92)

where $f' = f'(\tilde{W}_1 \langle X\rangle)$ and $J = \tilde{W}_1 \operatorname{diag}(f'(\tilde{W}_1 \langle X\rangle))$. The $G_2$ and $G_1$ that optimize value can be computed iterating

$$G_i^{k+1}(t_i) = G_i^k(t_i) + \alpha_g \frac{dV}{dG_i(t_i)} \tag{93}$$

using algorithm 1. The obtained $G_i(t)$ from this optimization process are plugged into $\hat{Y} = \tilde{W}_2 f\left(\tilde{W}_1 X\right)$, then trained using SGD to check how much of improvement we get using the computed control signal inferred using the approximated dynamics. The results for this model are depicted in Figure 20.

## J    DATASET DETAILS

**Correlated gaussians**: Toy dataset with correlated gaussian inputs. We sample $y_1$ as $\pm 1$ with probability $1/2$. Then sample $y_2 = y_1(1 - 2\xi)$ with $\xi \sim \operatorname{Ber}(p)$, if $p = 1/2$ then the labels are independent. We generate the input $x_i \sim \mathcal{N}(y_i\mu_i, \sigma_i^2)$, then taking $X = [x_1, x_2]^T$ and $Y = [y_1, y_2]^T$. From this data distribution process, we can analytically compute $\Sigma_x = \langle XX^T\rangle$, $\Sigma_{xy} = \langle XY^T\rangle$ as $\Sigma_y = \langle YY^T\rangle$

$$\Sigma_x = \begin{bmatrix} \mu_1^2 + \sigma_1^2 & \mu_1\mu_2(1 - 2p) \\ \mu_1\mu_2(1 - 2p) & \mu_1^2 + \sigma_1^2 \end{bmatrix}, \quad \Sigma_{xy} = \begin{bmatrix} \mu_1 & \mu_1(1 - 2p) \\ \mu_2(1 - 2p) & \mu_2 \end{bmatrix} \quad \text{and} \quad \Sigma_y = \begin{bmatrix} 1 & 1 - 2p \\ 1 - 2p & 1 \end{bmatrix}.$$
(94)

$\langle X\rangle = 0$ and $\langle Y\rangle = 0$.

**Hierarchical concepts**: This is a semantic learning dataset used in (Saxe et al., 2019; Braun et al., 2022) to study learning dynamics when learning a hierarchy of concepts. This task allows linear neural networks to present *rich* learning (opposite to *lazy* learning, see (Chizat et al., 2020; Flesch et al., 2022)) when using a small weight initialization. The *rich* regime shows a step-like learning dynamics where each step represents a learning of a different hierarchy level. In this dataset, $\Sigma_x$ and $\Sigma_{xy}$ have a close form, for example, for a hierarchy of 3 levels,

$$\Sigma_x = I_4, \quad \Sigma_{xy} = \begin{bmatrix} 1 & 1 & 1 & 1 \\ 1 & 1 & 0 & 0 \\ 0 & 0 & 1 & 1 \\ 1 & 0 & 0 & 0 \\ 0 & 1 & 0 & 0 \\ 0 & 0 & 1 & 0 \\ 0 & 0 & 0 & 1 \end{bmatrix} \quad \text{and} \quad \Sigma_y = \Sigma_{xy}^T \tilde{\Sigma}_{xy} \tag{95}$$

where $I_n$ is the identity matrix of size $n$. $\langle X\rangle_j = \frac{1}{I}\sum_{j=1}^I (\Sigma_x)_{ji}$ and $\langle Y\rangle_j = \frac{1}{I}\sum_{j=1}^I (\Sigma_{xy})_{ji}$. Samples from this dataset are simply $X$ as a random column from the identity matrix and $Y$ as the corresponding column from $\Sigma_{xy}$.

**MNIST**: This is a classification task of hand-written digits (Deng, 2012). Images from this dataset were reduced to $5 \times 5$ and flattened. We estimated the correlation matrices, $\hat{\Sigma}_x$, $\hat{\Sigma}_{xy}$ and $\hat{\Sigma}_y$ by taking the expectation over the samples in the training set.

# K ADDITIONAL RESULTS

Here we present extra Figures and discussion to some of the results from the main text.

## K.1 SINGLE NEURON MODEL

We did several runs varying some hyperparameters of the system to see the effect on the optimal control signal and the improvement in the instant reward rate $v(t)$.

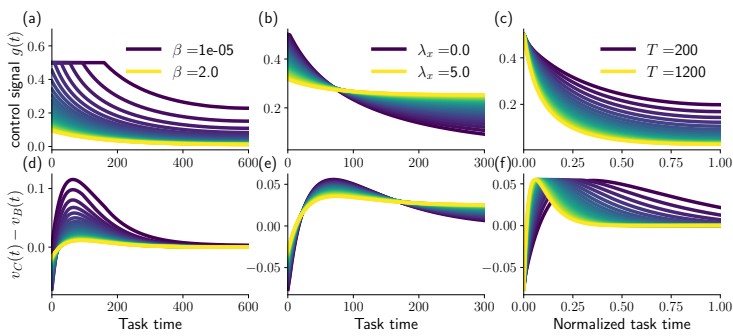

Figure 10: Results of hyperparameter variations for the single neuron case. **(a) and (d)**: optimal control signal $g(t)$ and difference between instant net reward using control $v_C(t)$ and the baseline (no control) net reward $v_B(t)$, for the variations of the cost coefficient $\beta$. Panels **(b) and (e)** and **(c) and (f)** are same results but varying regularization coefficient $\lambda$ and available time to learn $T$.

In Figure 10 we show the results for these runs varying the cost coefficient $\beta$ (Figures 10a and 10d), the regularization coefficient for the $L_2$ norm of the weights $\lambda$ (Figures 10b and 10e), and different available time to learn the task $T$ (Figures 10c and 10f). Increasing $\beta$ leads to less control. Note that for $\beta = 0$ the amount of control does not explode, since changing $g(t)$ after learning the task will alter the input-output mapping function, leading to an increase in the loss. Increasing $\lambda$ leads to an overall similar amount of control, but the control is sustained longer for higher $\lambda$, this is due to the high cost of increasing the size of the weight $w(t)$, which is absorbed by $g(t)$ to get closer the optimal solution $w^*$, by making $\tilde{w}(t) = w(t)(1 + g(t))$ closer to $w^*$. To allow comparison, we normalized the time axis for the variations of available time, this parameter does not seem to change the control signal or instant reward rate within the time span to learn the task.

## K.2 EFFORT ALLOCATION

Here we present results and analysis of the gain modulation model trained on single datasets. Figures 11 and 12 depict the results of the effort allocation using gain modulation trained on the gaussian and semantic datasets respectively. As mentioned in Section 6, in all cases the learning of the dataset is speed up by the learning effort control signal. Most of the control is exerted in early stages of learning (to compensate with reward in later high reward stages of training), with peaks around times when the improvement in the loss is higher due to the weight learning dynamics, decaying to zero at the end of the given time frame. The $L_2$ norm of the weights is roughly higher when using control throughout the training, but it converges to the same value for the baseline and control case. Different are the trajectories for the $L_1$ norm (measuring sparsity), where the set of weights gets more sparse when using control, to minimize the cost of using the control signal while keeping the effect of it over the weights still high. This can be explicitly seen when training in all of the dataset when inspecting the weights and control evolution through time, shown in Figures 13, 14 and 15 for the gaussian, semantic and MNIST datasets respectively. There is a cluster of weights that move closer to zero compared to the baseline training, and a few weights (near the number of non-zero gain coefficients in $G_i(t)$) become larger with time. Given that the input-output mapping is linear $\left(\hat{Y} = \tilde{W}_2(t)\tilde{W}_1(t)X\right)$, the solution for the linear regression problem (taking $\lambda = 0$ for simplicity) is given by $W^* = \tilde{W}_2\tilde{W}_1 = \Sigma_{xy}^T\Sigma_x^{-1}$ for both the baseline and the gain modulation case,

and both cases reach the global solution for the linear regression problem (Fig. 5c). In the controlled case, because the instant net reward $v(t)$ also considers the cost of having $G(t) \neq 0$, the purpose of the control is to change the learning dynamics and reach a solution such that $W^* = \tilde{W}_2 \tilde{W}_1$ and $G(t) = 0$. Since regularization is considered in the backpropagation dynamics, the weights for the baseline and controlled case reach the same $L_2$ norm, but the weights when optimizing control are in general more sparse, as shown in the $L_1$ norms throughout learning in Fig. 5b. The reason for this is the over-parametrized nature of the network, having more parameters $W$ than the ones needed to solve the linear regression.

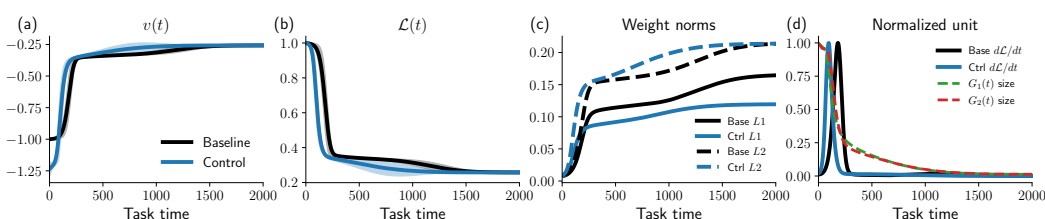

Figure 11: Results of the gain modulation model trained on the gaussian dataset. **(a)**: Instant net reward $v(t)$, baseline vs controlled. **(b)**: Loss $\mathcal{L}(t)$ throughout learning. **(c)**: L1 and L2 norms of the weights. **(d)**: normalized $d_t\mathcal{L}(t)$, and normalized L2 norm of the control signal $G_1(t)$ and $G_2(t)$.

In the particular case of the effort allocation model trained on the semantic dataset, we can see two other extra features. First, because we initialized the network with small weights ($\sim 10^{-4}$), we can see step-like transitions in the loss through time $\mathcal{L}(t)$, a regime known as *rich learning* (Chizat et al., 2020; Flesch et al., 2022), where each step corresponds to learning one of the hierarchical concepts in the dataset, from highest to lowest. In this regime, the control signal is able to skip the plateaus when learning each level on the hierarchy. In addition, the control signal is the highest just the first step and decays exponentially until the end of training. We infer that the effect on the dynamics by the control signal is not merely scaling the learning rate, but since each neuron has its own independent gain modulation, the control signal can guide the complex learning dynamics to avoid facing step-like transitions in the loss function.

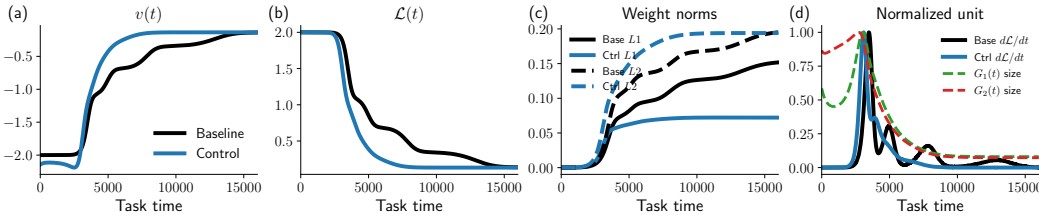

Figure 12: Results of the gain modulation model trained on the semantic dataset. **(a)**: Instant net reward $v(t)$, baseline vs controlled. **(b)**: Loss $\mathcal{L}(t)$ throughout learning. **(c)**: L1 and L2 norms of the weights. **(d)**: normalized $d_t\mathcal{L}(t)$, and normalized L2 norm of the control signal $G_1(t)$ and $G_2(t)$.

### K.3 TASK SWITCH

In Figures 16 and 17, additional results for the gain modulation model trained on the task switch are presented. In Figure 16, note that the weight norm for the controlled case through the switches are larger. The cost of switching is transferred to the weights by making them larger, so the use of the control signal is less costly when switching. This can also be seen in Figure 17, where the control signal is large only for a few weights, which in the long terms are the ones that become larger, reducing also the size of control needed to change the effective input-output transformation $\hat{Y} = \tilde{W}_2 \tilde{W}_1 X$. In addition, the weights when using control are grouped in two clusters, the ones influenced by the control signal which have larger absolute values, and the rest are pushed near zero

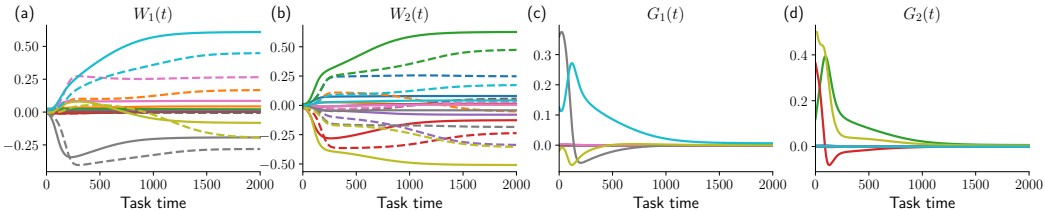

Figure 13: Weight and control signal evolution through training in the gaussian dataset using the effort allocation task from Section 6. **(a)**: First layer weights $W_1(t)$ (baseline and control training depicted with solid and dashed lines respectively). **(b)**: Second layer weights $W_2(t)$. **(c)**: First layer gain modulation $G_1(t)$. **(d)**: Second layer gain modulation $G_2(t)$. Each color corresponds to a specific weight, and colors match between plots of weights and the control signal.

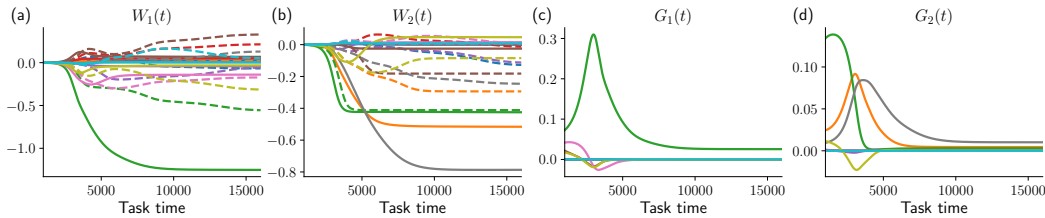

Figure 14: Weight and control signal evolution through training in the smantic dataset using the effort allocation task from Section 6. **(a)**: First layer weights $W_1(t)$ (baseline and control training depicted with solid and dashed lines respectively). **(b)**: Second layer weights $W_2(t)$. **(c)**: First layer gain modulation $G_1(t)$. **(d)**: Second layer gain modulation $G_2(t)$. Each color corresponds to a specific weight, and colors match between plots of weights and the control signal.

(opposite to what is seen in the baseline case, with weights spread around zero). The gain modulation allocates resources for every switch in only a few weights, while the rest of the weights are pushed closer to zero to avoid interfering with the inference process through when switching.

### K.4 TASK ENGAGEMENT

The set of datasets for the task engagement experiment was chosen based on how hard is to solve them with linear regression. In Figure 18a the best achievable loss when classifying MNIST digits using linear regression is depicted for pairs of digits. The pairs used in the task engagement experiments, $(0, 1)$, $(7, 1)$ and $(8, 9)$ have corresponding 0.02, 0.036, and 0.055 optimal loss $\mathcal{L}^*$ respectively, therefore ordered from easiest to harder according to this metric.

### K.5 CATEGORY ASSIMILATION

By taking the average across columns of the matrix in Figure 18a, we obtained the average minimum loss per digit when compared to any other digit, shown in Figure 18b (color bar with normalized values). When training the engagement modulation on the category assimilation task (Results in Section 5), the order of learning numbers is roughly the same as the difficulty in terms of linear separability. The easiest digits are focused on first, then harder ones. According to the linear separability metric, the order from easier to hardest are 0, 6, 1, 4, 7, 9, 2, 3, 8, 5, similar to what is depicted in Figure 4.

In addition to the engagement modulation model trained on the category assimilation task, we trained a restricted gain modulation model using the *neuron base* as described in Appendix H.1.1. In this model, the gain modulation for the first layer is disabled, then the only extra parameter adjusted to maximize expected return in equation 2 are the coefficients of the base $\nu_2^b$. These coefficients scale

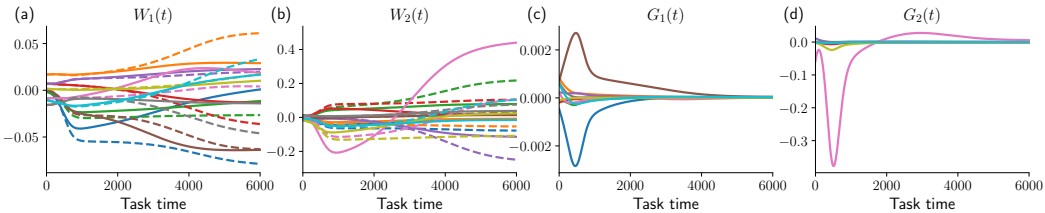

Figure 15: Weight and control signal evolution through training in the MNIST dataset using the effort allocation task from Section 6. **(a)**: First layer weights $W_1(t)$ (baseline and control training depicted with solid and dashed lines respectively). **(b)**: Second layer weights $W_2(t)$. **(c)**: First layer gain modulation $G_1(t)$. **(d)**: Second layer gain modulation $G_2(t)$. Each color corresponds to a specific weight, and colors match between plots of weights and the control signal.

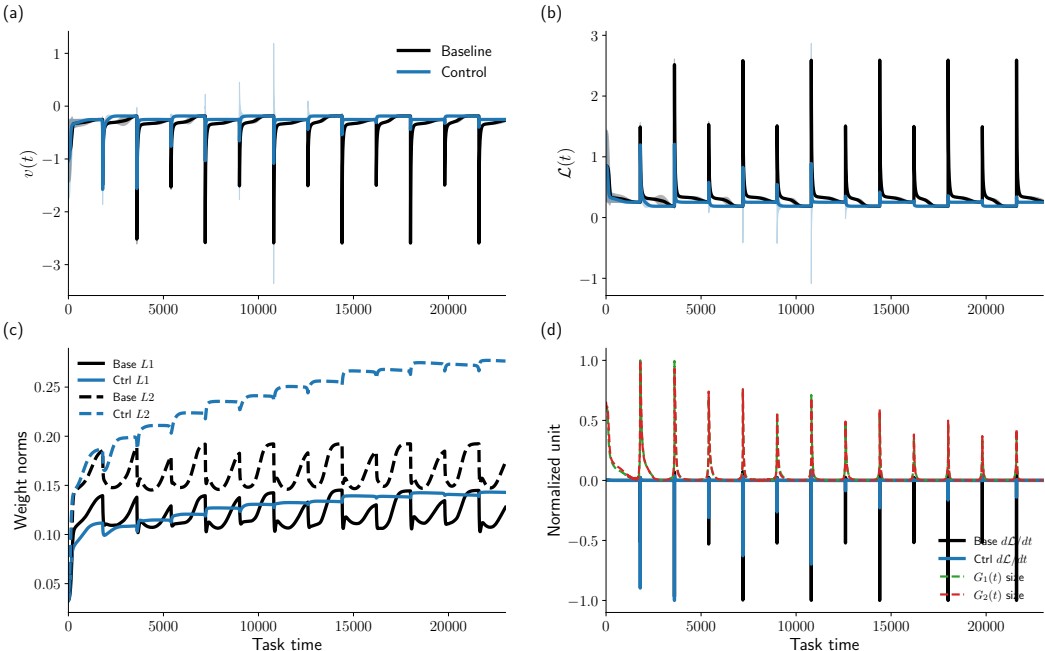

Figure 16: Additional results of the gain modulation model trained on the Task Switch. **(a)**: Instant net reward $v(t)$, baseline vs controlled. **(b)**: Loss $\mathcal{L}(t)$ throughout learning. **(c)**: L1 and L2 norms of the weights. **(d)**: normalized $d_t\mathcal{L}(t)$, and normalized L2 norm of the control signal $G_1(t)$ and $G_2(t)$.

the response of output neurons in the second layer, scaling the error signal, but also the value of the error signal itself (since it is changing the mapping as well). The results are similar in terms of order of the order of digits engaged through learning as shown in Figure 19, but the effect in the loss function is smaller because of the influence on the error signal (the value itself, not the scaling). This effect can be made explicit when deriving the learning dynamics equations for the weights (taking $G_1(t) = 0$) (for example for $W_2$), giving

$$\tau_w \frac{dW_2}{dt} = \left[\left(\Sigma_{xy}^T - \tilde{W}_2 W_1 \Sigma_x\right) W_1^T\right] \circ \tilde{G}_2(t) \tag{96}$$

$$= \nu \left[\left(\underbrace{\Sigma_{xy}^T - \nu W_2 W_1 \Sigma_x}_{\text{error signal}}\right) W_1^T\right] \tag{97}$$

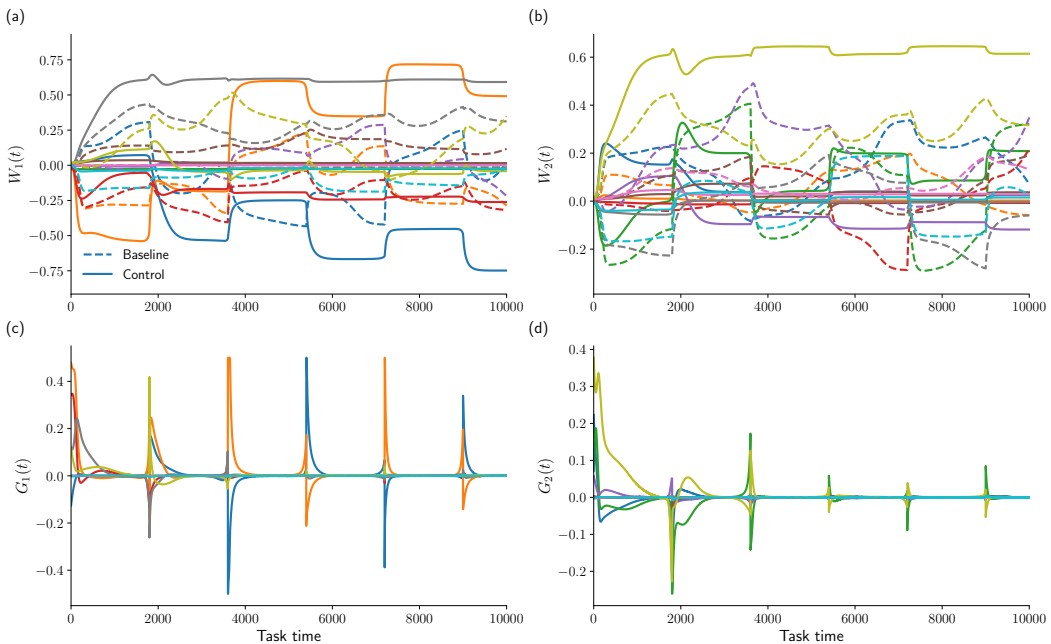

Figure 17: Weight and control signal evolution through training in Task Switches. **(a)**: First layer weights $W_1(t)$. **(b)**: Second layer weights $W_2(t)$. **(c)**: First layer gain modulation $G_1(t)$. **(d)**: Second layer gain modulation $G_2(t)$. Each color corresponds to a specific weight, and colors match between plots of weights and the control signal.

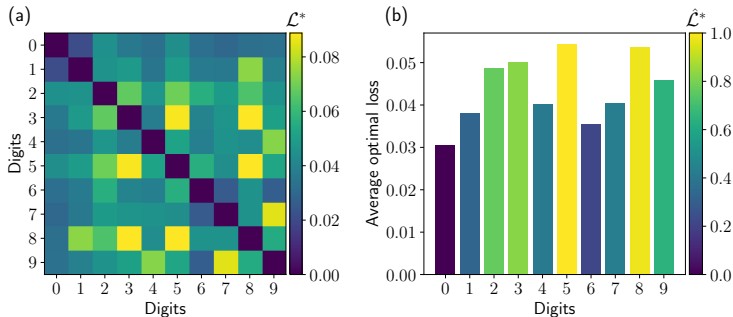

Figure 18: **(a)**: Minimum error achievable $\mathcal{L}^*$ when classifying between 2 digits (rows and columns) using a linear classifier. **(b)**: Average of minimum error achievable per digit across all digits (average per row), color bar shows this normalized quantity.

where $\nu$ is a diagonal matrix with the coefficients $\nu_2^b(t)$ in the diagonal. Both the *learning rate per output* (the $\nu$ outside the square parenthesis) and the error signal are influenced by these coefficients. In the case of the engagement modulation explained in section H.3, the coefficients just scale the error signal in equation 83.

## K.6 NON-LINEAR NETWORK

### K.6.1 DYNAMICS APPROXIMATION

We used the gain modulation model (Effort Allocation experiment) to train a non-linear network on the gaussian dataset. We used a Taylor expansion around the mean to obtain an approximated

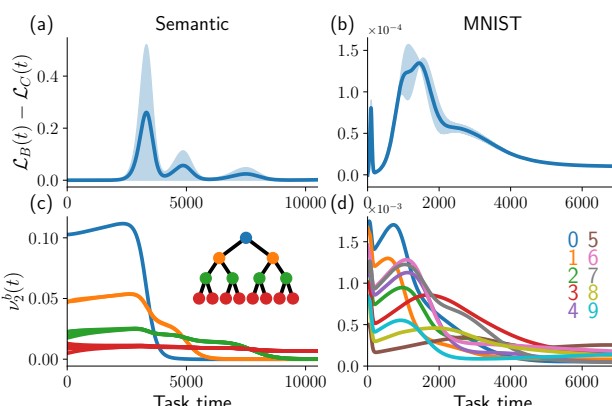

Figure 19: Results for category assimilation task using the *neuron base* when restricting the gain modulation model. **(a) and (b)**: Improvement in the loss function when using control for MNIST and Semantic dataset respectively. **(c) and (d)**: Optimal category engagement coefficients for MNIST and Semantic respectively.

equation for the non-linear dynatmics as explained in Appendix I. Because the non-linear function chosen is $\tanh(\cdot)$, and we initialize the network using small weights, the learning dynamics of the non-linear network are near linear at the beginning of the training, so the estimated weights and the real ones from SGD training are close as shown in Figure 20 (panels (c), (d), (g), (h)). This is useful to estimate the control signal since most of the control is exerted in the early stages of learning, as depicted in Figures 20i and 20j (and in all linear networks testes throughout this work). The obtained control signal using the approximated dynamics is still able to improve training of the real non-linear network using SGD and gain modulation as shown in Figures 20a and 20b.

### K.6.2 NON-LINEAR NETWORK VS LINEAR NETWORK

To show the capacity of linear network dynamics to describe the dynamics of non-linear networks, we trained a non-linear network using stochastic gradient descent by sampling batches on the semantic task, we then optimized the category engagement control signal as in Section 5 and derived in App. H.3. The parameters are exactly the same as in the results shown in Figure 4(h and i), except in this case the dynamics are described by batch training, and the first layer has a non-linear function $\tanh(\cdot)$. These results suggest that **linear network dynamics resembles the dynamics of non-linear networks training, as well as the optimal meta-learned control signal**. First, note that the step-like evolution in the loss function is described by both, the linear and non-linear network training (Figure 4(h and i) and Figure 21a). Second, the control signal (engagement per category in the semantic task $\phi_c(t)$) can improve learning in this non-linear network, in the same way as in the linear network case. Third, the optimal engagement per attribute shown in Figure 21b follows the same qualitative behavior as in the linear case. Attributes higher in the hierarchy in the semantic task are focused first compared with lower ones in the hierarchy. Harder attributes are engaged later but with more sustained effort in time. These are all found also in the linear network case as shown in the results of Section 5. Both networks scale linearly in time with the number of iterations in the inner loop dynamics, but the non-linear network optimization takes an order of magnitude more. In terms of memory cost, the training for the non-linear network increases linearly with batch size. Increasing the batch size facilitates the training of the meta-variables since it reduces the variance of the estimation in the gradient, then obtaining a trade-off between the stability of the meta-optimization and batch size when meta-optimizing control in a non-linear network by batch training. Further analysis should be done to characterize the computational costs of each case.

On the other hand, writing closed-form ODEs available in linear networks (and the work referenced in App.A and B), allows for a compact description of the entire training of the network by using a few carefully chosen summary statistics, instead of saving every data point. This is why the meta-learned control signal acting on the ODEs allows for some mathematical and computational tractability.

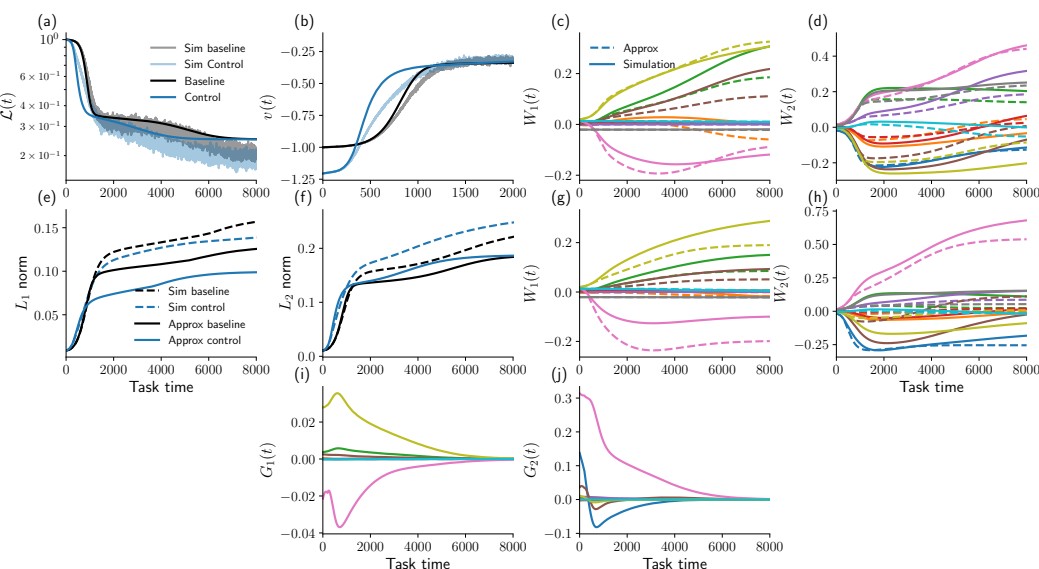

Figure 20: Results of the learning effort framework in a non-linear network using a linear approximation. **(a) and (b)**: $\mathcal{L}(t)$ and $v(t)$ respectively. Solid lines are numerical solutions to the approximated learning dynamic for the baseline case in equation 91 and the control case in equation 92. Simulation training the real non-linear network using SGD shown in shaded lines, using the inferred gain modulation in the real non-linear network when using control. **(c) and (d)**: $W_1(t)$ and $W_2(t)$ respectively. In the baseline training, comparing the numerical solution of the approximated dynamics in equation 91 (solid lines) with the weights from the simulation in the real non-linear network trained using SGD (dashed lines). **(e) and (f)**: $L_1$ and $L_2$ norms of the weights through training. Solid lines are numerical solutions to the approximated learning dynamic for the baseline case in equation 91 and the control case in equation 92. Simulation training the real non-linear network using SGD shown in dashed lines, using the inferred gain modulation in the real non-linear network when using control. **(g) and (h)**: $W_1(t)$ and $W_2(t)$ respectively. In the control case training, comparing the numerical solution of the approximated dynamics in equation 92 (solid lines) with the weights from the simulation in the real non-linear network trained using SGD and using the gain modulation computed to maximize expected return (dashed lines). **(i) and (j)**: Control signals for first and second layer $G_1(t)$ and $G_2(t)$ respectively.

Our goal is to show that this is a promising direction by exhibiting a large number of settings we can accommodate with this framework while keeping some of the analytical tools. We believe that this will motivate further mathematical analysis and techniques addressing the dynamics of optimal meta-learning.

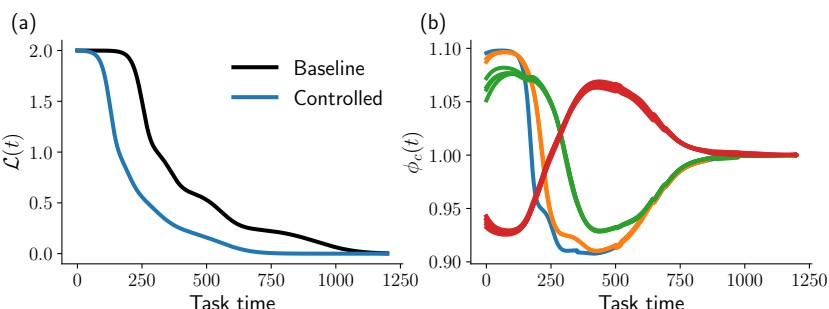

Figure 21: Category engagement modulation in a non-linear network. Results here were obtained with the same parameters as the one used for Figure 4(h and i). (a) Step-like learning dynamics of the non-linear network, and optimized dynamics using category engagement $\phi_c(t)$ to maximize value $V$. (b) Optimal category engagement per level in the hierarchy (as in Figure 4(h and i)).

## L  Simulation Parameters

Table 1: Optimization Parameters for the single neuron example in Section 2.1, shown in Figures 2 and 10. Every other variable is constant when varying each particular value of the parameter sweep.

| Parameters | Notation | Value |
|---|---|---|
| Network learning rate | $\alpha$ | 0.001 |
| Regularization coefficient | $\lambda$ | 0.1 |
| Control lower bound | $g_{\text{MIN}}$ | 0 |
| Control upper bound | $g_{\text{MAX}}$ | 0.5 |
| Discount factor | $\gamma$ | 0.99 |
| Reward conversion | $\eta$ | 1.0 |
| Control cost coefficient | $\beta$ | 0.3 |
| Control learning rate | $\alpha_g$ | 10.0 |
| Control gradient updates | $K$ | 700 |
| Weight time scale | $\tau_w$ | 1.0 |
| Available time (A.U.) | $T$ | 600 |
| Mean of Gaussians | $\mu$ | 2.0 |
| Intrinsic noise | $\sigma_x$ | 1.0 |
| Batch size | B | 128 |
| $\gamma$ values sweep | 10 ** (NP.LINSPACE(-8, 0, NUM=30, ENDPOINT=TRUE)) | |
| $\beta$ values sweep | NP.LINSPACE(1E-5, 2, NUM=30, ENDPOINT=TRUE) | |
| $\sigma_x$ values sweep | NP.LINSPACE(1E-5, 5, NUM=30, ENDPOINT=TRUE) | |
| $\lambda$ values sweep | NP.LINSPACE(0, 5, NUM=30, ENDPOINT=TRUE) | |
| $T$ values sweep | [200 + I*50 FOR I IN RANGE(21)] | |

Table 2: Parameters used for MAML simulation in App. F.1. The distribution of tasks where 5 MNIST pairs, (0, 1), (7, 1), (8, 9), (3, 8) and (5, 3), see App. J. Weights initialized from a gaussian distribution centered at 0 with standard deviation of 0.01

| Parameters | Notation | Value |
|---|---|---|
| Network learning rate | $\alpha$ | 0.005 |
| Hidden units | $H$ | 40 |
| Regularization coefficient | $\lambda$ | 0 |
| Control lower bound | $g_{\text{MIN}}$ | not bounded |
| Control upper bound | $g_{\text{MAX}}$ | not bounded |
| Discount factor | $\gamma$ | 1.0 |
| Reward conversion | $\eta$ | 1.0 |
| Control cost coefficient | $\beta$ | 0 |
| Control learning rate | $\alpha_g$ | 0.005 with ADAM |
| Control gradient updates | $K$ | 2000 |
| Weight time scale | $\tau_w$ | 1.0 |
| Optimized steps | $T$ | from 1 to 340 every 20 |

Table 3: Parameters used for learning rate $\alpha(t)$ optimization in App. F.2. $\gamma$ and $\beta$ where varied independently, keeping the default values when varying the other.

| Parameters | Notation | Value |
|---|---|---|
| Network learning rate | $\alpha$ | 0.005 |
| Hidden units | $H$ | 40 |
| Regularization coefficient | $\lambda$ | 0 |
| Control lower bound | $g_{\text{MIN}}$ | -1 |
| Control upper bound | $g_{\text{MAX}}$ | 1 |
| Reward conversion | $\eta$ | 1.0 |
| Control learning rate | $\alpha_g$ | 0.005 with ADAM |
| Control gradient updates | $K$ | 800 |
| Weight time scale | $\tau_w$ | 1.0 |
| Available time (A.U.) | $T$ | 18000 |
| default discount factor | $\gamma$ | 1.0 |
| default cost coefficient | $\beta$ | 1.0 |
| $\gamma$ values sweep | NP.LINSPACE(0.6, 1.0, NUM=10, ENDPOINT=TRUE) | |
| $\beta$ values sweep | NP.LINSPACE(1E-3, 2, NUM=10, ENDPOINT=TRUE) | |

Table 4: Optimization Parameters for results in Section 5. Dataset parameters are the following: For the **Attentive, Active and Vector** models, the three datasets used are MNIST digits, being $(0, 1)$, $(7, 1)$ and $(8, 9)$, each a binary classification task with a batch size of 256, images reshaped to $(5 \times 5)$ and flattened. **Eng. MNIST**: all digits were used with a batch size of 256, images reshaped to $(5 \times 5)$ and flattened. **Eng. Semantic**: Batch size of 32 and 4 hierarchy levels described in J. For every dataset, an extra 1 was concatenated to the input to account for the bias when multiplying by the weights.

| PARAMETERS | NOTATION | ATTENTIVE | ACTIVE | VECTOR | ENG MNIST | ENG SEMANTIC |
|---|---|---|---|---|---|---|
| NETWORK LEARNING RATE | $\alpha$ | 0.005 | 0.005 | 0.005 | 0.05 | 0.005 |
| HIDDEN UNITS | $H$ | 20 | 20 | 20 | 50 | 30 |
| REGULARIZATION COEFFICIENT | $\lambda$ | 0.0 | 0.0 | 0.0 | 0.0 | 0.0 |
| ENG. COEF. LOWER BOUND | $\psi_{\text{MIN}}, \phi_{\text{MIN}}$ | 0.0 | 0.0 | 0.0 | 0.0 | 0.0 |
| ENG. COEF. UPPER BOUND | $\psi_{\text{MAX}}, \phi_{\text{MAX}}$ | 2.0 | 1.0 | 2.0 | 2.0 | 2.0 |
| DISCOUNT FACTOR | $\gamma$ | 0.99 | 0.99 | 0.99 | 0.99 | 0.99 |
| REWARD CONVERSION | $\eta$ | 1.0 | 1.0 | 1.0 | 1.0 | 1.0 |
| CONTROL COST COEFFICIENT | $\beta$ | 0.1 | 0.1 | 0.1 | 5.0 | 5.0 |
| CONTROL LEARNING RATE | $\alpha_g$ | 1.0 | 1.0 | 1.0 | 1.0 | 1.0 |
| CONTROL GRADIENT UPDATES | $K$ | 800 | 800 | 800 | 600 | 600 |
| WEIGHT TIME SCALE | $\tau_w$ | 1.0 | 1.0 | 1.0 | 1.0 | 1.0 |
| AVAILABLE TIME (A.U.) | $T$ | 13000 | 13000 | 13000 | 30000 | 18000 |

Table 5: Optimization Parameters for results in Section 6 and Appendix I. Dataset parameters are the following: **MNIST**: Batch size, 32; reshape size, $(5 \times 5)$; Digits, $(1, 3)$. **Gaussian and Non-linear**: Batch size, 32; $\mu_1 = 3$, $\mu_2 = 1$, $\sigma_1 = 1$, $\sigma_2 = 1$, $p = 0.8$. **Semantic**: Batch size, 32; hierarchy levels, 4. **Task Switch**: Gaussian 1: $\mu_1 = 3$, $\mu_2 = 1$, $\sigma_1 = 1$, $\sigma_2 = 1$, $p = 0.8$; Gaussian 2: $\mu_1 = -2$, $\mu_2 = 2$, $\sigma_1 = 1$, $\sigma_2 = 1$, $p = 0.2$, switch every 1800 iterations. For every dataset, an extra 1 was concatenated to the input to account for the bias when multiplying by the weights.

| PARAMETERS | NOTATION | MNIST | GAUSSIANS | SEMANTIC | TASK SWITCH | NON-LINEAR |
|---|---|---|---|---|---|---|
| NETWORK LEARNING RATE | $\alpha$ | 0.005 | 0.005 | 0.005 | 0.005 | 0.001 |
| HIDDEN UNITS | $H$ | 50 | 6 | 30 | 8 | 8 |
| REGULARIZATION COEFFICIENT | $\lambda$ | 0.01 | 0.01 | 0.01 | 0.001 | 0.0 |
| CONTROL LOWER BOUND | $g_{\text{MIN}}$ | -0.5 | -0.5 | -0.5 | -0.5 | -0.5 |
| CONTROL UPPER BOUND | $g_{\text{MIN}}$ | 0.5 | 0.5 | 0.5 | 0.5 | 0.5 |
| DISCOUNT FACTOR | $\gamma$ | 0.99 | 0.99 | 0.99 | 0.99 | 0.99 |
| REWARD CONVERSION | $\eta$ | 1.0 | 1.0 | 1.0 | 1.0 | 1.0 |
| CONTROL COST COEFFICIENT | $\beta$ | 0.3 | 0.3 | 0.3 | 0.3 | 0.3 |
| CONTROL LEARNING RATE | $\alpha_g$ | 10.0 | 10.0 | 10.0 | 1.0 | 10.0 |
| CONTROL GRADIENT UPDATES | $K$ | 1000 | 1000 | 1000 | 1000 | 500 |
| WEIGHT TIME SCALE | $\tau_w$ | 1.0 | 1.0 | 1.0 | 1.0 | 1.0 |
| AVAILABLE TIME (A.U.) | $T$ | 16000 | 16000 | 16000 | 23000 | 10000 |
| RESULTS | | FIGURE 5 | FIGURE 11 | FIGURE 12 | FIGURE 5 | FIGURE 20 |

