# OpenReview forum: "Meta-Learning Strategies through Value Maximization in Neural Networks"
_ICLR.cc/2024/Conference — Submitted to ICLR 2024_

### Official Review · Reviewer_QJtu · 2023-10-26

**Soundness:** 4 excellent
**Presentation:** 4 excellent
**Contribution:** 3 good
**Rating:** 8
**Confidence:** 2

**Summary:**

The paper explores various meta-algorithmic choices ("control signals") that influence the learning dynamics of neural networks. The authors define optimal control signals by the condition that they maximize the cumulative learning performance (minus control costs) over the learning process. They study the setting of deep linear networks which allows analytic solutions of the learning dynamics equations, and, consequently, enables gradient-based optimization of the control signal. In the experiments, the authors consider control signals describing parameter initialization, learning-rate choice, learning curriculum, and gain modulation. The authors discuss the relevance of their results in the context of cognitive control in neuroscience.

**Strengths:**

The writing style of the paper is excellent and the authors supplement their exposition with helpful explanations and figures (e.g., Fig. 1). The proposed framework appears as a nice approach to reason about meta-algorithmic choices in the cognitive sciences and machine learning. The authors derive a general framework and apply it to a range of experiments which seem to be well fleshed out. I appreciate that the authors study an analytically tractable variant (linear networks) and derive results that seem to be consistent with the cognitive control literature. While I am uncertain of the practical relevance of the experimental results (cf. weaknesses), I consider theoretical studies of the proposed kind interesting and relevant in their own right. Thus, I recommend acceptance (with low confidence).

**Weaknesses:**

While the authors argue that their results are mostly consistent with the cognitive control literature (on which I cannot comment because I'm not an expert in this field), I doubt that the results have any practical relevance for the design of modern machine learning/meta-learning algorithms. I would be interested in whether the authors think that any practical advice for the machine learning practitioner can be derived  from their experiments.

**Questions:**

cf. weaknesses

---

> ### Author Response · Authors · 2023-11-18
>
> Thank you for your review. We are glad you found our writing style excellent, and our experiments well explained. We note that our work offers a path to get intuition on meta-learning strategies, and hopefully motivate further theoretical work on this tractable setting. Please, see below our response to your comments.
>
> > Relevance for modern machine learning models and practical advice
>
> To show relevance in more complex models, we added a new experiment using a non-linear neural network trained on the semantic task, to then optimize the category engagement control signal to maximize value using our framework. The results of this new experiment show that the learning dynamics of linear networks resemble the ones from a non-linear network, including the optimal meta-learning signal found in this non-linear network (See qEL8 for more details).
>
> Regarding practical advice to train modern architectures, we can suggest a few things from the conclusions of our work. For example, we recover the general idea of learning rate annealing which is generally used in modern architectures (See Figure 8 in Appendix F.2). This was not forced into the meta-learning system in any way, and just comes out from finding the optimal learning rate signal to maximize value through the learning trajectory. Another example is the idea of a curriculum from easy to hard examples. Besides the links with cognitive neuroscience mentioned in App. B, there is literature on machine learning developing tools around this concept (Bengio, et al. (2009). “Curriculum learning”). In Sinha et al. 2021 “Curriculum by smoothing”, the authors blur the feature maps of a convolutional neural network (large and complex architecture) to expose the network to lower frequency components on images at the early stages of the training, while exposing the network to higher frequency components in latest stages of learning. This is a way of curriculum where lower frequency information is arguably easier, and higher frequencies are harder. Exposing the network to easy components in early stages improves learning as we also concluded using our framework. We hope more of these types of interventions can be discovered, or perhaps, fully understood in simplified settings or through mathematical analysis. We will add these examples in the extended discussion. Thank you for this question.

---

> > ### Comment · Reviewer_QJtu · 2023-11-22
> > **Thank you for the comment**
> >
> > Thanks for your answer. Although the other reviewers tend to disagree with my assessment, I still consider the paper an interesting contribution that is presented well and provides a nice approach to reason about meta-algorithmic choices. Personally, I think such papers can be valuable for the community, even if their practical applicability might be limited. Therefore, I keep my score.

---

### Official Review · Reviewer_LGyh · 2023-11-01

**Soundness:** 3 good
**Presentation:** 4 excellent
**Contribution:** 1 poor
**Rating:** 6
**Confidence:** 3

**Summary:**

In this paper, the authors present a meta-learning framework based on control signals via discounted cumulative performance.
In this framework, the authors describe a model, learning dynamics and a discounted reward based on a measure of the performance.
The authors then present a simplifying example that helps the reader better understand the approach and then continues with a two layer linear network model, evaluating on different perspectives of a classification task.
The authors show how this approach generalizes other existing meta-learning methods and present different instances of meta-learning tasks.

**Strengths:**

The paper is technically sound and presents a great exposition that helps the reader better understand the ideas in the paper.
The experiments are backed by an extensive appendix that clarifies details.

**Weaknesses:**

The main limitation of the paper is as the authors mention, based on the linear models they use. This limits applicability.
Another limitation is the lack of comparison against other meta-learning instances, where the evaluation could compare computational time. However, the linear limitation probably makes this a non-important issue, and lifting it might introduce tractability problems.

**Questions:**

I found it interesting how the control adapts as expected. However, I'm wondering if for the given linear models that you present, if we were to collapse the two layers into a single one, your meta-learning turns into a form of regularization or instance weighting?

---

> ### Author Response · Authors · 2023-11-18
>
> Thanks for your review. We appreciate you found our paper well explained and with an extensive description of the framework and experiments. The goal of our simplified setting is to give a better intuition on how the framework is implemented, we are glad you found it useful. Please, find our answers to each of your comments below:
>
> > Non-linear networks
>
> We further justify our use of linear networks by providing an extra experiment on a real non-approximated non-linear network, showing we obtain similar results as when using a linear network (see answer to reviewer qEL8).
>
> > Relation to other meta-learning algorithms
>
> We find the connection between our framework and standard meta-learning algorithms very important. This is why we show that our framework can instantiate MAML, and bilevel programming (App F1 and F2). This is not a comparison but a way to show that they are mathematically equivalent under certain parameter configurations. Describing this connection is a way to motivate the use of our framework to understand meta-learning solutions to other problems as well.
>
> We discussed the applicability of scalable meta-learning algorithms to our setting in Appendix F, second paragraph. Scalable meta-learning algorithms that can be applied to state-of-the-art architectures are either simplifying the meta objective (compared to the value function computed in our framework), restricting the optimized meta-variables, or approximating some of the relevant quantities (like meta-gradients). As we mentioned in this discussion, “each step in our outer loop considers the entire inner loop learning trajectory and computes the gradient of the meta-loss at all time steps to capture the effect of the meta-parameters across the entire learning trajectory, not just the last step as in the referenced work. In addition, our meta-parameters can be as complex as the (size of the network) times (inner loop training iterations), in other words, our meta-variables also depend on time, increasing the complexity of the optimization problem. We are solving the full complex meta-learning problem (which is the desired target in both references), by considering a simpler model, instead of approximating our computation.” This can be seen explicitly in App. C where we show the form of the meta-gradient to update the control signal. Further discussion of other meta-learning algorithms we are not covering with our framework can be found in Appendix F4. In addition, we included a new experiment in Appendix F3 where we meta-learn the learning rule, further showing the capabilities of our framework to relate to other meta-learning problems.
>
> > Collapsing the two layers into one:
>
> This is an interesting question, and the answer reflects the complexity that linear network dynamics is bringing to our model.
>
> First, you can collapse the forward path to a single matrix, as
> $$ \hat{Y} = W_{2}W_{1}X = WX$$
> because the network is linear. The best solution we can find using this network is linear regression, but the path it takes from the initial weights to the solution is given by the non-linear dynamics of $W_1$ and $W_2$, so it does not follow the same trajectory as a linear regression trained using gradient descent, due to the weight coupling between $W_{1}$ and $W_{2}$. The next step we could do is to compute gradient descent updating $W$ directly. We included the derivation of the learning dynamics of $W$ under a control signal which can be written in closed form (see Appendix G2). Each $W(t)$ depends on every previous value of the control signal g, giving a non-trivial relation that does not look like a common regularization (it might regularize in some way that is not explicit, such as finding sparser solutions as shown in the results of Section 6). The reason for this is that $W(t)$ reduces the loss instantly, while g reduces the cumulated loss. This can be also seen in the computation of the gradient of the control signal in App. C. Since we are optimizing the control signal in the gradient flow regime, there is no such thing as “single samples” from the dataset in the dynamics, but rather the summarized statistics of the data as $\Sigma_{x}$, $\Sigma_{xy}$ and $\Sigma_{y}$, hence instance weighting does not apply.

---

> > ### Comment · Reviewer_LGyh · 2023-11-23
> >
> > I thank the authors for the clarifications and I will raise my score

---

### Official Review · Reviewer_eAk8 · 2023-11-01

**Soundness:** 3 good
**Presentation:** 3 good
**Contribution:** 3 good
**Rating:** 5
**Confidence:** 2

**Summary:**

The paper presents a framework for meta-learning where the learning dynamic can be analytically solved. The framework optimizes the control signals through gradient descent by considering  the total discounted learning performance minus control costs as an objective. Analytical tractability is reached by using a simple two-layer linear neural network that simplifies the dynamic. The authors show that they can model MAML, Bilevel Programming and other meta-learning methods with the introduced framework.

**Strengths:**

1) By suitable choice of the control signal, one can model MAML, Bilevel Programming, task switch, and other techniques.
2) The experimental section is vast, well-described, and explained.

**Weaknesses:**

The authors propose the framework as a test bed for meta-learning. However, I have several concerns regarding its practical applicability:

1) A simple two-layer linear neural network is used, so it may not account for all the effects during training, and the results may not translate to more complex NNs.
2) It's likely one will need to find a solution for every novel considered intervention.
3) I don’t think all interventions can be modeled within this framework, even when restricted to the context of two-layer linear neural networks. (please see the Question section)

So I believe the framework may have limited usage as a test bed.

**Questions:**

1) Will the framework be able to find tractable solutions if we consider learning rules as a control signal (e.g. [1])?
2) To what extent can we generalize control signals in a two-layer linear neural network while still maintaining solution tractability?
3) Do you think it is possible to develop task switching in your framework when one doesn't know what task will be next in advance?

Small incorrectness.
Images (b) and (c) should be swapped In Fig. 2.

[1] Andrychowicz, Marcin, et al. "Learning to learn by gradient descent by gradient descent."

---

> ### Author Response · Authors · 2023-11-18
> **Answer Summary**
>
> Thanks for your review. We are glad you found the experimental section vast and well explained. As we mentioned in other responses, our framework can accommodate a large set of interventions, mostly in the context of cognitive neuroscience, but we believe it might help the study of meta-learning strategies in machine-learning models as well. In this work, we have shown at least 10 (+ 2 more in the revised paper) different combinations of learning settings and interventions, describing them thoroughly. Our goal is to show that many more interventions can be studied using this framework, for example, the one you suggest on learning the learning rules [1]. We have added an experiment in this direction. We describe more details below.

---

> ### Author Response · Authors · 2023-11-18
> **Detailed Answer**
>
> > W1: Use of linear networks and comparison to non-linear networks.
>
> As mentioned in the answer to reviewer qEL8, the dynamics of linear networks are non-linear with a non-convex landscape, resembling the dynamics of more complex non-linear neural networks. The part that is linear is the forward path, but the control signal is dealing with essentially a non-linear dynamical system due to the nonlinear loss function and weight coupling between the first and second layers as described in Sec 3. We run an additional experiment to show this, please see the answer to reviewer qEL8.
>
> > W2: It's likely one will need to find a solution for every novel considered intervention.
>
> We agree with your statement. However, as shown in the appendix for each of the models, the math necessary to write the differential equations describing learning as a function of the control signal, in most cases, is standard gradient computation. Your intuition is right, changing the intervention of the control signal or the learning system, will change these equations. This is part of the challenge when extending to non-linear networks. Techniques to describe learning dynamics in non-linear networks exist (see paragraph 3, appendix A), but they have their own drawbacks in terms of mathematical and computational tractability, or required limits in input or hidden dimensionality to obtain closed-form differential equations describing the dynamics. This is a direction we are currently exploring and we can definitely see a path forward, but this is beyond the scope of the paper, since the general framework (Sec 2) presented here is the same.
>
> > W3, Q1 and Q2: Other interventions and learning rules as a control signal
>
> As you mentioned in your review, the experiments presented in this work are vast, and due to the flexibility of our framework, we do not believe it is possible to cover all possible interventions in a single manuscript. The tractability of each control intervention will largely depend on the learning system, data distribution, and degrees of freedom of the control signal. This is all a design choice, for example, in App. H.1.1 we discussed the possibility of using a low-rank basis for the control signal, which might simplify computation and search space. We acknowledge that not all meta-learning phenomena can be instantiated with our framework. There is a literature in meta-learning that we are not covering with our framework, where there is no explicit variable that is meta-optimized. We discussed this in App. F4.
>
> Thank you for the suggestion, using learning rules as control signals is possible, and we now include an experiment demonstrating one approach. In particular, we have considered the setting of “Cao, Summerfield, and Saxe et al. 2020, NeurIPS”, which describes a space of possible learning rules using two parameters, spanning gradient descent, Hebbian contrastive learning, quasi-predictive coding, Hebbian and anti-Hebbian rules. These rules instantiate several popular ideas in theoretical neuroscience. Concretely, for the deep linear network setting, the weight updates are
>
> $$ \Delta W_{1} = \Delta W_{1}^{C}(\gamma) + \Delta W_{1}^{H}(\eta), \hspace{0.2cm} \Delta W_{2} = \Delta W_{2}^{C}(\gamma),$$
>
> with $\Delta W_{i}^{C}$ being a contrastive learning rule, and $\Delta W_{i}^{H}$ a Hebbian learning rule, which are controlled by the parameters $\gamma$ and $\eta$ respectively.
> Using our framework, we can find the optimal $\gamma(t)$ and $\eta(t)$ that maximizes value as described in Sec 2. By inferring these parameters, the system is deciding which learning rule to employ within this space over learning. We found that the optimal learning rule is consistent with the original Cao et al. 2020 paper. We provide a new experiment in App F3 which will be referenced in the main text. These methods could offer a route to examining how a learning agent might adapt its learning algorithm to better learn different tasks, for instance. We believe this additional experiment further shows the flexibility of our framework to accommodate different meta-learning settings.
>
> > W3: Switching without knowing the next task in advance
>
> This is an important future direction. We note that in many cognitive science experiments, task switches happen at predictable times (after every 50 trials, say), and our analysis is directly applicable to this setting. To consider the case where task switches are unpredictable would require extensions: fundamentally, one could compute the expected loss over possible next tasks, either analytically in certain cases or through sampling. In essence, while we roll out a single learning trajectory corresponding to a predictable sequence of tasks, a sampling approach would roll out many trajectories with possible next tasks, and construct the loss function as the sum of these tasks. The control signal would then be optimized to perform well for the distribution of switches.

---

### Official Review · Reviewer_qEL8 · 2023-11-02

**Soundness:** 3 good
**Presentation:** 3 good
**Contribution:** 3 good
**Rating:** 6
**Confidence:** 3

**Summary:**

The works introduces a learning effort framework which is capable of optimizing control signals on objectives with discounted cumulative performance throughout learning. Frameworks and settings being analyzed in the work are meta-learning, curriculum learning, and continual learning and a number of results are provided, showing how and when the control effort might be helpful under linear settings.

**Strengths:**

1. The authors present a learning effort framework over a number of problem settings to analyze optimal strategies for learning. Having an understanding and intuition about this can help our current deep learning design problems learn better.
2. The document motivates the problem well, emphasizing the importance of the work.
3. The arguments made by the work are linked to cognitive science and neuroscience, which can be used to get inspiration from when designing our current models.
4. The work conducts experiments over multiple paradigms, including meta-learning & continual learning.

**Weaknesses:**

Some of the areas that the work could be improved upon:
1. As pointed out by the authors themselves, a limitation is the assumption of linear models. Since the motivation behind the current work is to provide ways in improving the current neural networks, more analysis on non-linear systems is needed. Although very large neural networks are hard to analyze, simpler variations could be considered for non-linear settings to make this direction even more interesting.
2. Optimization is a big challenge in neural networks. An analysis focused on the effect of different optimization techniques might be helpful when extending the work on non-linear settings, which is currently being analyzed in only simpler artificial settings.
3. A number of other biases exist in the cognitive literature, eg having modular systems in which different modules do different tasks, or learn them over multiple tasks. Analyzing this explicitly can be an interesting addition to the work.
4. Figure 1 could be made a bit larger for better readability.

**Questions:**

1. Have the authors come across any unexpected results while analyzing the settings? For example, for curriculum learning where defining the curriculum itself might effect the learning in unexpected ways.
2. Is it possible to include some experiments from non-linear models (without the approximation) to address the concerns from the "weaknesses" section?

---

> ### Author Response · Authors · 2023-11-18
> **Answer Summary**
>
> Thanks for your thoughtful review of our paper. We are pleased that you found our paper is well-motivated, offers useful intuition, contains many experiments in diverse settings, and usefully highlights links to cognitive neuroscience, which is a main purpose of our paper. Our goal is to show the flexibility of the framework to account for other learning settings that go beyond the experiments we presented here. Regarding your comments, while we agree that the use of linear networks is a limitation of the current paper, it is not an in-principle limitation of the framework itself. As in other parts of deep learning theory, we start with linearity as a useful prerequisite before tackling nonlinear extensions, and emphasize that deep linear networks still yield a nonconvex loss landscape and nonlinear dynamics that bears similarity to nonlinear networks. We further support the relevance of our deep linear network analysis by running our framework on a non-linear network, showing that similar qualitative results hold for non-linear networks. We also indicate the path forward to extensions of this framework to include non-linear network dynamics.

---

> > ### Author Response · Authors · 2023-11-18
> > **Detailed Answer**
> >
> > > W1 and W2: New experiment, use of linear networks, and extension to non-linear networks.
> >
> > While the forward path of the networks used in this paper is linear, the gradient descent dynamics on the weights of multi-layer linear networks is non-linear if we think of them as a dynamical learning system. As we mentioned in Section 3, the dynamics are non-linear due to the nonlinear loss function, weight coupling and non-convex loss landscape. In other words, the optimized control signal is dealing with a non-linear system (the learning dynamics of the linear network). As also mentioned in App. A and B, a linear network’s non-linear dynamics resemble the dynamics of other more complex non-linear networks (see paragraph 3, App. A). To show this qualitative resemblance, and thus the relevance of our linear analysis, we included a new simulation of a non-linear neural network trained on the category engagement modulation experiment described in Sec. 5 (new results in K.6.2). The results for this non-linear neural network resemble the ones of the linear neural network, in terms of learning dynamics and optimal meta-learning control found. While this nonlinear experiment is feasible in this small-scale setting, we note that the computational cost increases, whereas our framework remains tractable.
> >
> > We agree that the extension to non-linear networks is necessary, and we indicate related work to accomplish this in App. A and B. It is possible to just use batch training on a non-linear network to get the dynamics and optimize the control signal using our framework (as in the new experiment), but that requires storing all activations for the entire inner loop training period to do one gradient step on the control signal, as well as sacrificing mathematical tractability. On the other hand, writing closed-form ODEs available in linear networks (see referenced in App. A and B), allows for a compact description of the entire training of the network by using a few carefully chosen summary statistics, instead of saving every data point. Our goal is to show that this is a promising direction by exhibiting a large number of settings we can accommodate with this framework while keeping some of the analytical tools. We believe that this will motivate further mathematical analysis and techniques addressing the dynamics of optimal meta-learning. We included this discussion in App. K.6.2.
> >
> > > W3: Modularity and applications in cognitive neuroscience
> >
> > We first want to note that suggestions about applying our work to other problems are what we are looking for from the community with this paper. Other reviewers also came up with their applications which can be accommodated using our framework. We hope this comes as an advantage rather than a weakness. In this work, we focused on relevant applications for the cognitive neuroscience literature, but we think this might help the development of machine learning models as well (see App. A and B).
> >
> > Modularity is a question we are currently exploring, in terms of connectivity patterns between neurons and the representation of stimuli and tasks under different contexts. We believe it will be possible to consider cognitive control signals that activate or deactivate different modules to perform related tasks. This notion is present in classical models of cognitive control (Miller & Cohen, 2001), and the necessary ODE dynamics have been worked out in, for example, gated linear networks (Saxe et al. 2022). A range of continual learning questions could then be posed about when to modularize or not (Musslick et al. 2020, Sec 3.3.4, and Ravi et al. 2020). What our framework brings to the table is fewer assumptions underlying the analysis (we optimize the exact temporally discounted performance measure over fully time-dependent control signals), while keeping the complexity of a non-linear learning system and tractability. Further design decisions need to be made to account for modularity, such as architecture initialization, task/stimulus similarity, learning rules and time-scales, etc. This is an exciting direction we are currently taking using our learning effort framework, but it requires further description that goes beyond the scope of the already substantial number of settings considered in this paper.
> >
> > > W4: Figure 1
> >
> > Thank you, we will increase the size of the figure.
> >
> > > Q1:
> >
> > One surprise has been that we can find all of these intuitions from just maximizing value. This was not obligatory, as it could have been that certain intuitions arose from other aspects or considerations. We would also note that, while with the benefit of hindsight, the resulting control signals frequently are interpretable, before running these experiments we were often uncertain of the outcomes.
> >
> > > Q2:
> >
> > Thank you for the suggestion, we now include non-approximate results in nonlinear networks. See the answers above.

---

> ### Comment · Reviewer_qEL8 · 2023-11-22
> **Thank you for your responses for the rebuttal.**
>
> Thank you for clarifying the concerns raised, including concerns from other reviewers.
> I am updating my scores and will continue updating them after proof-reading the document again to see if all comments, including from other reviewers, have been addressed.

---

> > ### Author Response · Authors · 2023-11-23
> >
> > Thank you for engaging in the review process. Our rebuttal diligently addresses the concerns raised by the reviewers, providing new experiments that specifically address the use cases requested. We have also extended our discussions to offer comprehensive clarification on various questions raised.
> >
> > The reviewers have acknowledged the vast and well-explained applications of our framework as presented in the paper. Additionally, they have contributed their own use cases in this review, which we have successfully simulated, further enriching the potential impact of our work. We are confident that the cognitive neuroscience and machine learning community will explore their own applications of our framework.
> >
> > We acknowledge that we are not able to cover every application or connection to other concepts in a single paper. Having said this, in addition to the experiments presented, we have demonstrated the mathematical relationships with well-known meta-learning algorithms in machine learning (MAML and Bilevel Programming) and with the Expected Value of Control theory (a well-established theory in cognitive neuroscience). This highlights connections between ideas from machine learning and cognitive neuroscience in a computationally and mathematically tractable setting. With this, we think we have provided a solid foundation and justification for our framework.
> >
> > Thank you.

---

### Author Response · Authors · 2023-11-21

Thank you for your useful reviews which will greatly improve the quality of our paper and highlight the contribution of our work. We will be happy to further address any concerns you have about the paper. We're pleased to note that no technical errors were identified, and reviewers thought the experiments were 'vast' and well-explained.

A common concern was the extent to which our findings will improve modern machine learning systems. We note that we submitted to the neuroscience track, and that neuroscience & cognitive science applications are listed in the ICLR relevant topics list. We'd like to highlight the valuable implications our work holds for cognitive neuroscience, particularly for cognitive control, learning curricula, and attention, further underlining the broad-reaching impact of our research through several applications of our framework. That said, we do think our results could aid the design of modern machine learning systems. To support this claim, we presented 2 new experiments (in addition to the at least 10 experiments presented in the paper): First, one addressing the concerns of applicability of our framework on non-linear networks, showing that the results hold for linear and non-linear networks in the semantic tasks (App. K.6.2). Second, we used our framework to learn optimal learning rules (App. F.3), which relevant in the field of neuroscience and machine learning. As mentioned by the reviewers, we covered a vast set of experiments, and we think the community will come up with more applications of this framework.

We hope we can convey these ideas in the answers below. For more details, please refer to the specific answers of each reviewer. In particular, see our answers to reviewers qEL8 and eAk8 for a description of new experiments. We updated the pdf with new figures to facilitate our explanations.

Thank you.

---

### Comment · Area_Chair_3z33 · 2023-11-22

Dear reviewers,

This a reminder that deadline of author/reviewer discussion is AOE Nov 22nd (today). Please engage in the discussion, check if your concerns are addressed, and make potential adjustments to the rating and reviews.

Thank you!
AC

---

### Meta-Review · Area_Chair_3z33 · 2023-12-06

**Metareview:**

This work presents a general meta learning framework called "learning effort framework", which aims to systematically study optimal meta learning strategies in biological and artificial agents. The framework optimizes cumulative performance over control signals and can express MAML and Bilevel Programming by setting the framework's configurations. The paper also draws connections to the cognitive neuroscience literature and relates the framework to task engagement, cognitive control etc. The reviews recognize the technical contribution and nice presentation. However, the reviewers share common concerns on the limited practicality and contribution to the algorithmic meta learning community in machine learning. While the authors added new results for learning optimal learning rules and non-linear networks in the semantic tasks, several reviewers expressed their concerns that those experiments are still toy examples instead of real world applications (despite the fact that they gave ratings higher than 5).

While the primary area of the paper is "applications to neuroscience & cognitive science", the main focus of this paper is still a meta learning framework, which has some connections to neuroscience, but the exact *application to neuroscience & cognitive science* is unclear. In fact, the main *application* seems to be the design of machine learning systems. So I wonder whether a better characterization of this work is using inspirations from cognitive neuroscience to study meta learning in the machine learning area. Then the technical contributions do not seem to be enough since the empirical evidences on benefits to machine learning are too limited.

One more comment from Reviewer eAk8 which I think could be useful for the authors. (This comment did not play a role in the final decision)
> I believe the objective (2) (the integral over the future loss) was proposed many times before. So while reading the paper I imagine myself reading the educational textbook on meta-learning or neuroscience that illustrate the core idea of the meta-learning (objective (2)) with toy (but still hard to derive) examples. I like these examples, and I see the value of the paper for educational reasons, but I can't see how it can advance further research in these directions.

And when I asked for reference, the reviewer replied

>I don't have something specific in mind. But if one looks at RL, the value function estimates the future discounted rewards, and the agent is optimized to select actions that maximize a value function. So one can replace rewards with a loss to see similarity with the objective in the paper. Although in usual RL a value function estimates reward till the end of the episode. But in Meta-RL (https://arxiv.org/pdf/1611.02779.pdf) the value function is estimated across several episodes, or in continual RL there are no episodes, making it closer to the objective in the paper.

I believe this paper does have a lot of value and the connections to neuroscience are quite neat. But because of the lack of application to neuroscience & cognitive science and lack of contribution to ML, I recommend to reject this paper at ICLR.

**Justification For Why Not Higher Score:**

The reviewers share common concerns on the limited practicality and contribution to the algorithmic meta learning community in machine learning. While the authors added new results for learning optimal learning rules and non-linear networks in the semantic tasks, several reviewers expressed their concerns that those experiments are still toy examples instead of real world applications (despite the fact that they gave ratings higher than 5).

While the primary area of the paper is "applications to neuroscience & cognitive science", the main focus of this paper is still a meta learning framework, which has some connections to neuroscience, but the exact application to neuroscience & cognitive science is unclear. In fact, the main application seems to be the design of machine learning systems. So I wonder whether a better characterization of this work is using inspirations from cognitive neuroscience to study meta learning in the machine learning area. Then the technical contributions do not seem to be enough since the empirical evidences on benefits to machine learning are too limited.

**Justification For Why Not Lower Score:**

n/a

---

### Decision · Program_Chairs · 2024-01-16

Reject